# Path integration maintains spatial periodicity of grid cell firing in a 1D circular track

Pierre-Yves Jacob[1], Fabrizio Capitano[1], Bruno Poucet [1], Etienne Save[1] & Francesca Sargolini[1]

Entorhinal grid cells are thought to provide a 2D spatial metric of the environment. In this study we demonstrate that in a familiar 1D circular track (i.e., a continuous space) grid cells display a novel 1D equidistant firing pattern based on integrated distance rather than travelled distance or time. In addition, field spacing is increased compared to a 2D open field, probably due to a reduced access to the visual cue in the track. This metrical modification is accompanied by a change in LFP theta oscillations, but no change in intrinsic grid cell rhythmicity, or firing activity of entorhinal speed and head-direction cells. These results suggest that in a 1D circular space grid cell spatial selectivity is shaped by path integration processes, while grid scale relies on external information.

[1] Aix Marseille Université, CNRS, LNC UMR 7291, 13331 Marseille, France. Correspondence and requests for materials should be addressed to P.-Y.J. (email: pierre-yves.jacob@univ-amu.fr)

To navigate successfully, mammals can use both external landmarks and/or idiothetic cues derived from self-motion information[1,2]. Path integration is a navigational strategy based on idiothetic cues that requires the animal to estimate the distance and orientation relative to a starting location[3]. Based on their firing properties, grid cells in the medial entorhinal cortex (MEC) have been hypothesized to represent the neural substrate of path integration. Grid cells display a striking hexagonal grid-like firing pattern within an open field[4]. Their activity is modulated by running speed and heading direction suggesting that they integrate idiothetic cues to signal distance and direction information necessary for path integration[5–9]. Animal and human studies point to a role of the MEC in distance estimation[10–13]. However, how grid cells participate to such process and whether it is responsible for the grid cell periodic firing remain largely unknown. Distance can be calculated using external cues, self-motion information, or time elapsed[14]. From these different types of information, distance can be measured in four ways: (1) the allocentric distance based on external cues, (2) the path integrated distance, which is the distance referred to a fixed location and based on idiothetic cues, (3) the travelled distance, which is the summation of absolute distance travelled by the animal (also based on idiothetic cues), and finally (4) the distance measured by time elapsed[14]. Which information is used by the grid cell system to estimate distance has not been clearly identified so far. For example, in open-field tasks with different distal landmarks, grid cell activity is dominated by allocentric distance and path integration[15,16]. In contrast, when the animal runs on a treadmill (where the allocentric information is held constant and is hence irrelevant) time and travelled distance control grid cell activity[13]. Based on these studies it is not possible to distinguish between all possible computations, since either not all information types are available (as in the treadmill), or they cannot be easily separated (as in the open field). Moreover, since path integration requires the use of a space metric based on integrated distance[17], we would expect grid cells to process specifically this type of information.

In this study, we examined whether grid cell activity preferentially correlates with allocentric distance, path integrated distance, travelled distance, or elapsed time, in rats running in a continuous 1D environment, which allowed to disentangle the relative weight of the different coding mechanisms. Grid cells were recorded while rats were freely moving in a circular track (i.e. they were not trained to run unidirectional laps in the track, and could move at different speeds, either clockwise or counterclockwise), thus crossing the same location repeatedly and performing several laps during the same recording session (Fig. 1a). The circular wall of the track was uniformly black except for a white cue card attached to the external wall that helped polarizing the environment. If grid cells were coding allocentric distance, we would expect them to fire at the same position relative to the room cues over successive laps. If grid cells were coding distance based on path integration (i.e. path integrated distance), we would expect them to display firing fields that are regularly spaced across different laps. Accordingly, grid cell firing would not be anchored to the room cues, but rather would use each field as the spatial reference for the next one. If grid cells were coding distance based on the animal path (i.e. travelled distance), we would expect them to fire regularly according to the cumulative travelled distance regardless of the rat's position in the track. Finally, if grid cells were coding distance based on time (i.e. time distance), we would expect them to fire on a constant time step (Fig. 1b–e).

## Results

**Grid cell activity is aligned with the track**. To identify grid cells, MEC single units were first recorded while rats freely foraged in a 1.5-meter-diameter arena polarized by a white cue card attached to the wall. Of 365 well-separated MEC units, 64 were identified as robust grid cells recorded across six rats (Supplementary Figure 1, Table 1). Following recording in the circular arena, rats were confined to the 15-cm-wide peripheral rim of the arena by inserting an opaque inner wall. Although movements were restricted to the rim and therefore mostly one-dimensional, no barrier required the rat to turn about. Moreover, visibility of the cue card was limited to a region covering about 160° of arc in the circular track (supplementary information). This allowed us to analyze which coding process (allocentric, path integrated, travelled distance, or time) is used by grid cells to code the distance, and to investigate whether availability of the visual cue influences grid cell firing patterns. At the time of MEC recordings, all animals were highly familiar with both the arena and the track. Animals ran 3 laps in average, covering 42.3 m during 10 min and changing direction around 100 times in one recording session (Table 2).

The position of firing fields in the track did not correspond to the position of the fields in the arena (Fig. 2a, b). Distances between fields (i.e. field spacing) were highly irregular in the track, in contrast to the regular spacing observed in the arena. We first confirmed that field arrangement in the arena differed from that observed in the peripheral rim, by computing the correlation of grid cell activity in the corresponding spatial bins of the two setups (average correlation: $0.007 \pm 0.026$, not different from 0, one-sample $t$-test on Z scores, $t_{63} = 0.26$, n.s.), indicating that grid cell activity remapped in the track.

Grid cell activity in the track is unlikely to result from a slice through a 2D lattice based on the firing in the arena, as previously demonstrated[18–20]. For example, Yoon et al. (2016) showed that the firing fields of grid cells recorded in a linear track have non-periodic spacing and broad range of field heights. Moreover, the power spectral density (PSD) of their spatial activities revealed the presence of three dominant peaks, which is compatible with linear slices of a repeated 2D triangular pattern. None of these outcomes was found in our study. We observed that the range of field heights in the arena and the track were highly similar (Fig. 2c; paired $t$-test $t_{(63)} = 0.91$, n.s). Next, we showed that the PSD of grid cell firing in the track mostly presented one dominant peak (Supplementary Figure 2), which is compatible with a linear pattern of regularly spaced fields, rather than a 2D grid pattern. It also should be noted that grid cell activity in the track is not influenced by the running direction, in contrast to what is generally observed in a linear track[21,22]. The spatial maps from clockwise and counterclockwise trajectories were highly similar for all grid cells (average spatial correlation coefficient: $0.69 \pm 0.17$, one-sample $t$-test against 0, $t_{(63)} = 31.43$, $p < 0.001$). This result suggests that grid cell spatial code is different for linear and circular 1D spaces. In linear tracks, grid cell activity is controlled by both allocentric coding and path integration, similarly to 2D open environments, whereas in circular tracks path integration is mainly responsible for the grid cell spatial firing patterns.

We then asked whether grid cell activity in the track resulted from allocentric coding, i.e. if grid cells fired at the same position in the track on each lap (Fig. 1d-first row, Fig. 2d-top), or continuously integrated idiothetic information (Fig. 1d-second row). If the latter hypothesis is true, firing fields would precess on each lap, insofar as the field spacing is not an integer divider of the track circumference (Fig. 2d-middle). We found that grid cell firing fields are shifted from one lap to the next, suggesting that grid cell firing is not allocentric but involves path integrated distance computation (Fig. 2d-bottom). To quantify this effect, we first calculated spatial correlations between laps (Fig. 2e, Allocentric coding). Then, for each cell we calculated the average distance between firing fields, and we used that distance to shift

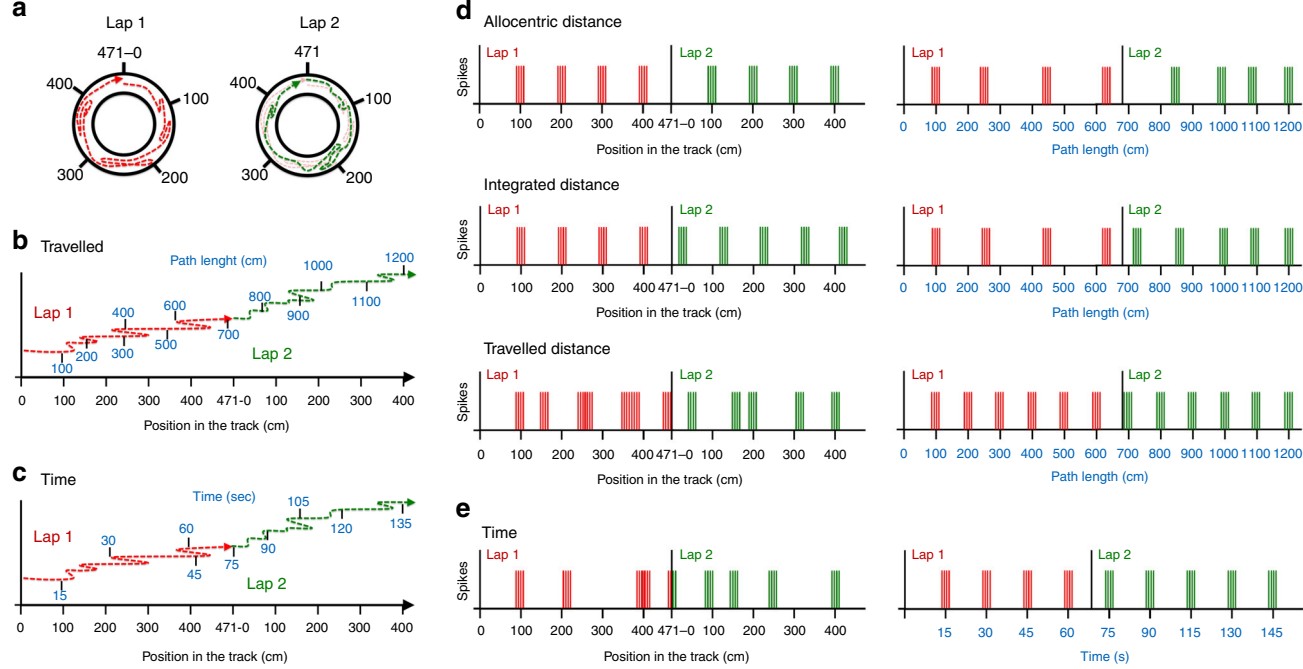

**Fig. 1** Predictions of grid cell firing based on allocentric distance, path integrated distance or travelled distance models. **a** Hypothetical trajectory of a rat in the circular track; red and green represent the first and the second lap, respectively. The total length of the track (471 cm) as well as cumulative 100 cm distances are indicated. **b**, **c** Hypothetical trajectory of a rat as a function of the position in the track. The blue values indicate the cumulative travelled distance (**b**) or the cumulative time elapsed (**c**), for each lap. The trajectories corresponding to the first and second laps are indicated in red and green, respectively. **d** Hypothetical position of grid cell firing fields relative to the track (left diagrams) or to the travelled path (right diagrams). Top row: according to the allocentric distance model, the firing fields keep the same position in the track across different laps (left), whereas their position relative to the cumulative travelled distance is highly variable from one lap to the other (right). Middle row: according to the path integrated distance model, the distance between the firing fields across laps is constant relative to the track (left), whereas their position relative to the animal path is highly irregular (right). Bottom row: according to the travelled distance model, firing fields are irregularly distributed around the track (left) but show a constant spacing across laps relative to the path travelled (right). **e** According to the time model, grid cells are expected to fire regularly based on a constant time step (right), and irregularly relative to the track positions (left)

**Table 1 Summary of the cell distribution**

|  | Number of recording session | Number of cells | Number of grid cells | % of grid cell |
|---|---|---|---|---|
| Rat 1 | 15 | 97 | 22 | 23 |
| Rat 2 | 12 | 91 | 17 | 19 |
| Rat 3 | 1 | 19 | 1 | 5 |
| Rat 4 | 5 | 48 | 7 | 15 |
| Rat 5 | 9 | 82 | 14 | 17 |
| Rat 6 | 3 | 28 | 3 | 11 |
| Total | 45 | 365 | 64 | Average = 15% |

**Table 2 Summary of behavior in the circular track**

|  | Average laps | Average travelled distance (meter) | Recording time (minute) | Number of turns |
|---|---|---|---|---|
| Rat 1 | 2.85 ± 1.05 | 47.8 ± 9.4 | 10 ± 1 | 84 ± 22 |
| Rat 2 | 1.98 ± 0.56 | 42.59 ± 3.74 | 9'30 ± 1'15 | 79 ± 10 |
| Rat 3 | 2.29 | 33 | 10'30 | 101 |
| Rat 4 | 3.91 ± 2.42 | 50.28 ± 15.1 | 14' ± 4 | 158 ± 50 |
| Rat 5 | 3.97 ± 1.05 | 33.41 ± 9.4 | 9'15 ± 2'50 | 109 ± 31 |
| Rat 6 | 2.15 ± 1.58 | 29.7 ± 7.10 | 8'52 ± 1'35 | 112 ± 10 |
| Total | 2.98 ± 1.2 | 42.3 ± 11 | 10 ± 2'30 | 98 ± 35 |

the firing map of each lap relative to the map of the previous lap (Supplementary Figure 3). If grid cells fire according to a path-integrated distance process, the distance correction will align the firing fields from lap to lap and consequently induce greater spatial correlation values (Fig. 2e, Path integrated coding). In agreement with this hypothesis, spatial correlation was significantly greater after the path integrated distance correction than in pure allocentric coding (Fig. 2e; paired $t$-test on Z scores: $t_{63} = -2.22$, $p < 0.05$). To control for a possible bias in this analysis, we computed the same spatial correlation analysis after splitting the path integrated firing map in two equal halves for 28 cells in which the animal ran at least 4 laps. We extracted the

average distance between firing fields from the first map, which we used to shift the laps of the second firing map. We confirmed that spatial correlation was significantly greater after the path-integrated distance correction than in pure allocentric coding (paired $t$-test on Z scores: $t_{28} = -5.89780$, $p < 0.001$).

We also checked whether spatial correlation after distance correction was significant for individual neurons. Of 64 grid cells, 50 (78.1%) yielded a significant correlation in the integrated distance model (i.e. shifted firing position from lap to lap) while only 11 (17.2%) yielded a significant correlation in the allocentric model (i.e. stable firing position from lap to lap). The remaining

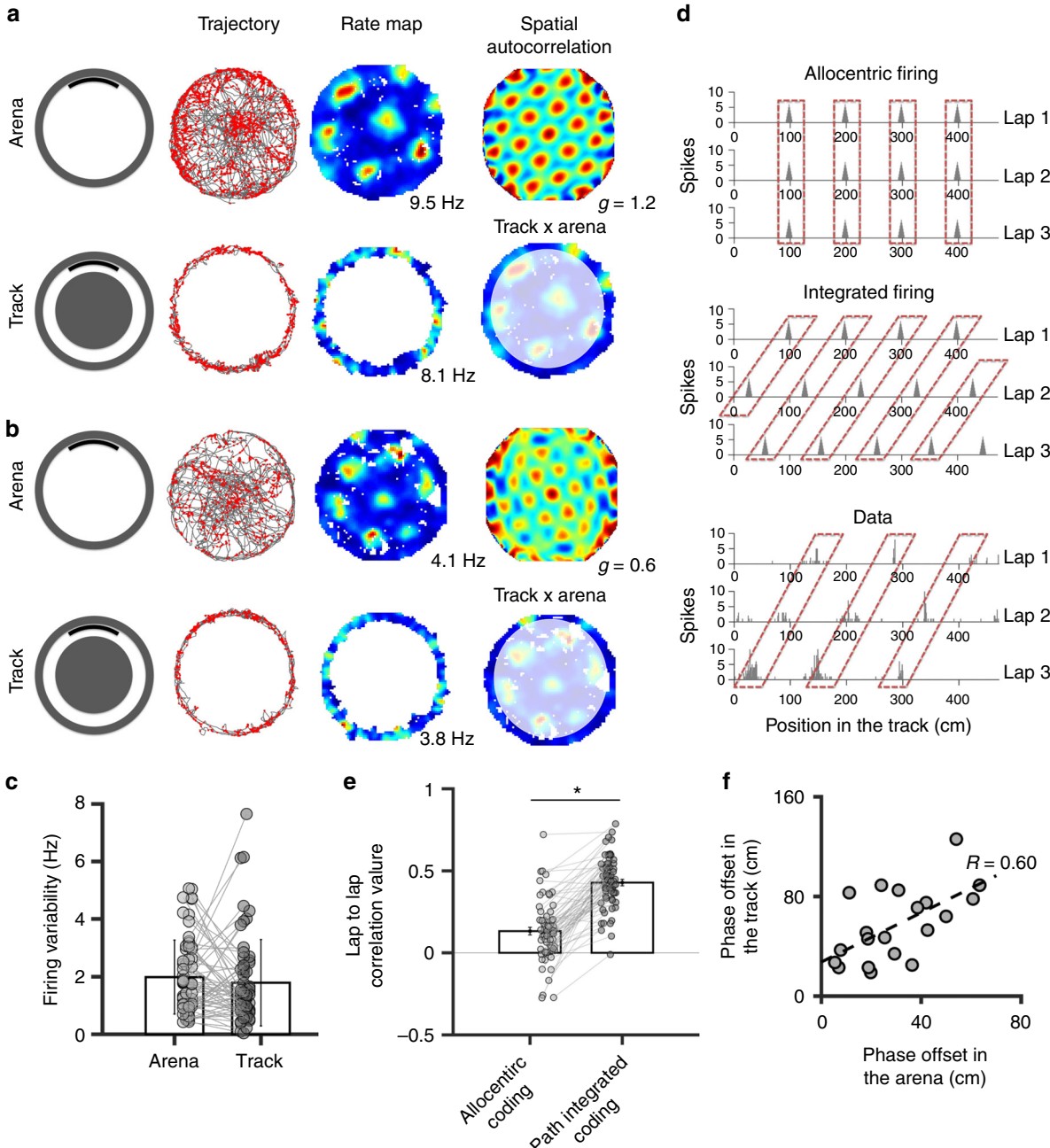

**Fig. 2** Grid cell activity is linearized in the track. **a**, **b** Activity of two grid cells recorded in the arena and in the track. Top row: from left to right, rat trajectory (in grey) with spike locations (red dots), color-coded rate maps (the peak of the firing rate is indicated) and 2D autocorrelograms. 'g' indicates the gridness score. Bottom row: from left to right, rat trajectory with spike locations, color-coded rate maps observed in the track, and rate maps in arena in which the external area (equivalent to the track) is highlighted. Note that there is no correspondence between the maps in the track and the peripheral area of the arena. **c** Average firing field variability (±SD) of grid cells in the arena and in the track. Data from individual cells are shown in grey. **d** diagrams showing rate plots of the activity of one hypothetical grid cell linearized in relation to the track position across three laps, according to the allocentric distance model (top, i.e. the position of the firing fields is stable across laps) and the path integrated distance model (middle, i.e. the position of the firing fields is shifted from one lap to the other). Bottom: activity from one grid cell recorded in the track; note that the position of the firing fields is consistent with the path integrated distance model. **e** Average correlation (±SEM) of grid cell activity across laps, according to the path integrated coding (i.e. the rate plot of one lap is shifted by a constant value with respect to the previous lap) and the allocentric coding (i.e. no shift between laps). Data from individual cells are shown in grey. *$p < 0.05$, paired $t$-test. **f** Scatter plots showing the relation between the phase offset of 20 pairs of simultaneously recorded grid cells in the arena and in the track. The Pearson correlation coefficient 'R' is indicated

three cells were not explained by either model. These results indicate that the firing of most grid cells defines a continuous regular pattern across different laps.

Despite such marked difference in the spatial codes, grid cell firing associations were maintained between the arena and the track, similarly to what was previously shown in different arenas[23]. We showed that the spatial phase shift between pairs of grid cells recorded in the arena and the track were highly correlated (Fig. 2f; correlation coefficient = 0.60, $p < 0.005$), suggesting that the pattern of co-activation is preserved across different environments, consistent with continuous attractor network models[17,24].

It should be noted that grid cells controlled by allocentric distance and grid cells controlled by path integrated distance were never recorded simultaneously. However, given the limited number of "allocentric" grid cells it is difficult to draw any firm conclusion from this observation.

**Path integrated distance shapes the linear grid cell pattern**. We then tested whether aligned grid cell firing is regular and whether this regularity better correlates with the path-integrated distance, the travelled distance, or elapsed time. We analyzed grid cell activity using two different methods.

In the first method, we linearized the path of the animal according to three reference frames: the track positions (i.e. path integrated distance), the distance travelled by the animal, or the time (Fig. 1d, e). Then, we averaged cell firing in 10 cm bins for spatial linearization (track and travelled reference frames) and in 1 s bins for time linearization, and we computed the autocorrelation of the firing activity for each linearized sequence. Results showed that grid cell firing produces more regular autocorrelation patterns for the path-integrated distance process than for both travelled and time processes (Fig. 3a, b). To test the significance of this regularity we used jitter control procedures for each coding scheme, and we extracted a threshold of correlation value equal to 0.18, 0.15, and 0.14 for path-integrated distance, travelled distance, and time, respectively (Fig. 3c–e). For each cell, we tested whether at least two peaks of the autocorrelation were higher than the threshold, and we found that the firing of 48 grid cells (75%) was correlated with path-integrated distance, whereas the firing of only one cell was correlated with travelled distance and none with time (Fig. 3c–e). A small proportion of grid cells (16 out of 64) did not show any significant modulation by path integration (see example in Fig. 3f–h). These cells showed significantly greater spatial correlation between laps compared to the path integration-modulated cells (Fig. 3i; paired $t$-test $t_{(63)} = 3.13$, $p < 0.005$), suggesting that allocentric coding was contributing to their firing pattern.

The second method consisted in fitting the observed data with simulations of firing activity according to path-integrated distance, travelled distance, and time models. We calculated a fitting score as the sum of square differences of the best fitting between the cell's activity and three parameters: the duration of spike trains, the distance (or time) of the first spike train, and the distance (or time) between spike trains. The path-integrated distance model yielded significantly greater fitting scores than travelled distance and time models and thus better explains grid cell activity (Fig. 4a; repeated measure ANOVA: $F_{2126} = 5.54$, $p < 0.005$; Tukey HSD post hoc, path integrated vs travel: $p < 0.01$, path integrated vs time: $p < 0.001$, travelled vs time: n.s.). We used best-fit parameters to reconstruct the firing rate map of each cell according to each of the three models, and we calculated the spatial correlation between these maps and the observed map (Fig. 4b, Supplementary Figure 4). Results confirmed that real maps correlate better with maps based on path integrated

distance (0.38 ± 0.18) than with maps based on travel model (0.21 ± 0.18) or time model (0.03 ± 0.18) (Fig. 4c; repeated measure ANOVA: $F_{2126} = 69.14$, $p < 0.001$; Tukey HSD post hoc, path integrated vs travel: $p < 0.001$, path integrated vs time: $p < 0.001$, travelled vs time: $p < 0.001$).

The two methods yielded consistent results since the distance values calculated were significantly correlated (correlation between distance values: 0.69, $p < 0.001$; Fig. 4d).

To sum up, grid cell spatial activity in a circular track is shaped by path integration and, to a minor extent, by allocentric coding. Neither travelled distance nor time contributed to the spatial firing patterns was observed in the track.

**Environmental features impact grid cell firing in the track**. The aforementioned results suggest that grid cell activity in the track mainly results from the integration of idiothetic cues from a fixed reference location. This computation possibly requires an interaction with static cues to correct idiothetic-based error accumulation[1,3]. Consequently, grid cell spatial selectivity in the track should decrease in the absence of visual cues. We tested this hypothesis by recording 35 grid cells in the track during light–dark–light sessions. Animal path was linearized according to the path-integrated distance model, and spatial correlation coefficients were calculated between light–dark and light–light conditions (Fig. 5a). Light–light average correlation yielded a mean value of 0.37 ± 0.21, while dark–light average correlation dropped to 0.01 ± 0.2 (Fig. 5b; paired $t$-test $t_{(34)} = 7.13$, $p < 0.001$). Consistent with these results, we observed that the activity of only 14/ 35 grid cells (40%) was explained by the integrated distance model in darkness, that is much less than the 48/64 grid cells (75%) in the light condition ($\chi^2_{(1)} = 11.72$, $p < 0.001$). These results indicate that idiothetic cues alone are not enough to maintain a regular firing pattern.

To further explore the influence of the visual cue, we checked whether firing stability was greater in the vicinity of the cue card. We first computed for each grid cell the autocorrelation of path-integrated firing in the region of the track in which the cue was visible. We performed the same analysis for the opposite region of the track in which the cue is not visible (Fig. 5c-left). We then compared the correlation values of the first peak of the autocorrelogram from the two regions and found that the average correlation was significantly greater in the visible-cue than in the non-visible-cue region of the track (Fig. 5c-right; paired $t$-test: $t_{(63)} = 4.82$, $p < 0.001$).

Next, we showed that, although mean firing activity was similar in both regions of the track (Supplementary Figure 5a; visible-cue region: 1.45 Hz ± 0.12; non-visible-cue region: 1.3 Hz ± 0.09; paired $t$-test $t_{(63)} = 1.53$, n.s), more number of spatial-fields were identified in the visible-cue region (233 fields, 65%) than in the non-visible-cue region (109 fields, 30%; $\chi^2_{(1)} = 23.24$, $p < 0.001$; Supplementary Figure 5b).

Altogether these results indicate that the availability of the visual cue strongly influences grid cell spatial firing. Furthermore, they suggest a possible role of the visual cue in correcting error accumulation over successive laps. Consistent with this hypothesis, grid cell spatial selectivity was not decreasing over time and remained stable within a recording session. We calculated the integrated distance firing for the first half and the second half of the recording session, and found no significant difference between the peak value of the autocorrelogram from the first half (0.39 ± 0.11) and the second half (0.33 ± 0.13) of the recording sessions (paired $t$-test: $t_{(63)} = 2.62$, n.s.).

Finally, we asked whether other environmental features might influence grid cell firing in the track. Studies have shown that field spacing is unmodified in familiar environments of different sizes,

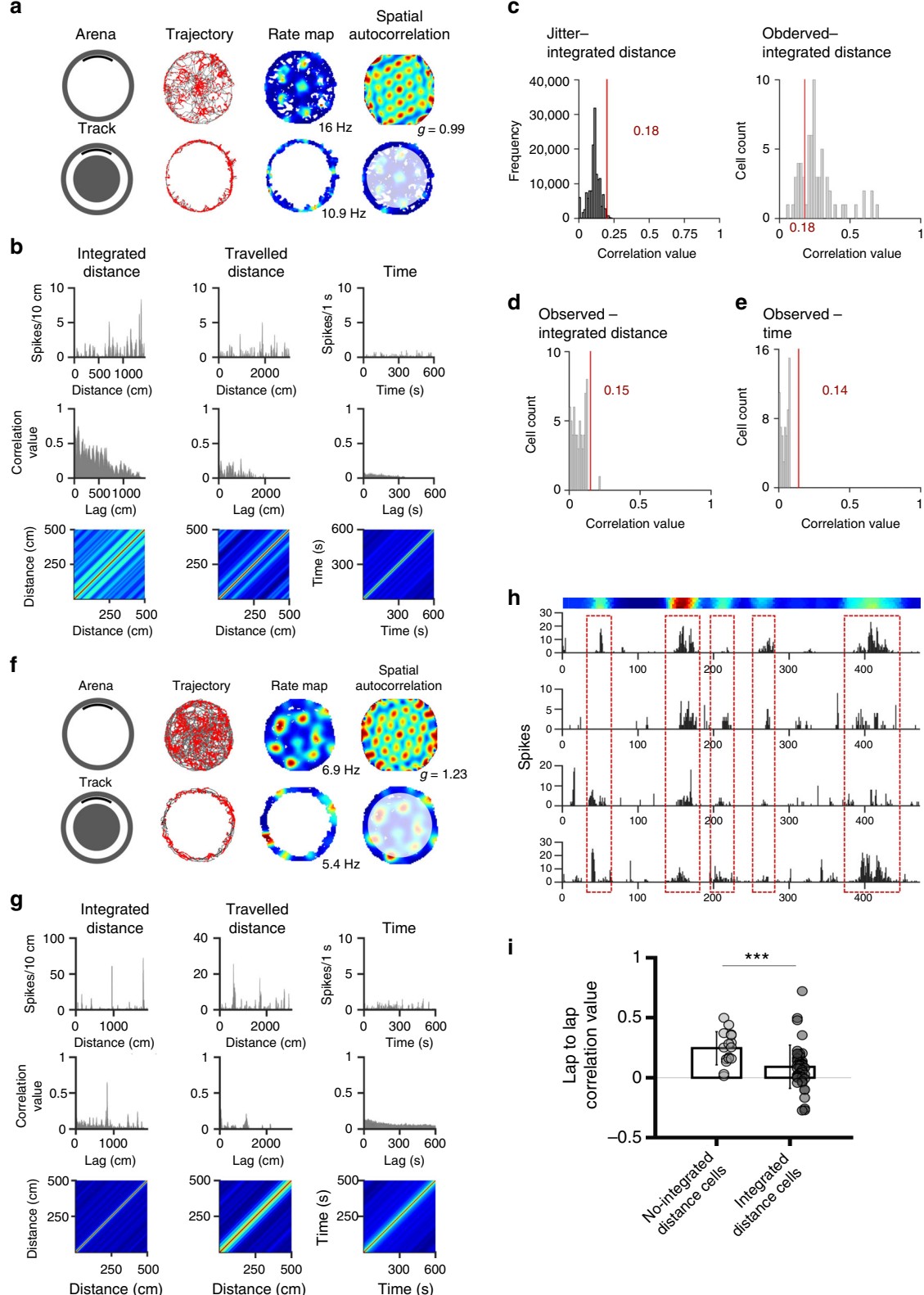

shapes, or contextual cues[4,23,25,26]. Thus, we recorded 34 grid cells in a small arena (∅100 cm) and its associated circular track as well as in the large arena (∅150 cm) and the large track (Supplementary Figure 5c-d). We found that in the small track, 29/34 cells (85%) had a firing pattern correlated with the path-integrated distance model, but no cell with either travelled distance or time, thus confirming the findings of the large track.

Furthermore, the proportion of grid cell firing explained by the path-integrated distance model was similar in the small and large tracks ($\chi^2_{(1)} = 1.39$, n.s.). As expected from previous findings[4,25,26] we observed similar field spacing in large and small arenas (spacing ratio = 0.97 ± 0.02). In contrast, we found that field spacing in the large track was 1.85 ± 0.57 greater than in the large arena and 1.55 ± 0.36 greater than in the small track.

**Fig. 3** Grid cell regularity is based on path integrated distance—autocorrelation analysis. **a** Activity of one grid cell recorded in the arena (top) and in the circular track (bottom), with peak firing rate and gridness score ('g'). **b** First row: activity of the grid cell (from panel a—track) linearized according to the path integrated distance (first column), the travelled distance (second column) and the time (third column). Second row: autocorrelation. Third row: Toeplitz matrix of the first 500 centimeters (corresponding to a complete lap). The color-coded matrix reveals regularity only for the path integrated distance. **c** Distribution of the average value of the first two peaks in the autocorrelation for all grid cells (Observed), as well as shuffled distribution obtained by resampling spike times from the same cells with a jitter procedure (Jitter; 400 permutations). Red lines indicate the ninety-ninth percentile for the shuffled data. **d, e** Distribution of the average value of the first two peaks in the autocorrelation of the linearized grid cell activity according to travelled distance (**d**) or time (**e**). Red lines indicate the ninety-ninth percentile for the shuffled data. Only one grid cell showed autocorrelation value above the statistical threshold, exclusively for the travel distance. **f–h** Example of one grid cell showing allocentric coding. **f** Activity of the cell in the arena (top) and the track (bottom). **g** Linearized firing activity (first row) of the cell according to the three distance models, autocorrelation (second row) and Toeplitz matrix. **h** Color-coded rate map of the cell activity linearized in relation to the track position (top row), and rate plots of the firing activity across four laps; note that the position of the firing fields is consistent with the allocentric firing model. **i** Average correlation (±SD) of grid cell activity across laps, according to the allocentric distance model (i.e. no shift between laps) for the 16 grid cells which do not show a path integrated distance firing and the 48 grid cells which show a path integrated distance firing. Data from individual cells are shown in grey. ***$p < 0.005$, paired $t$-test

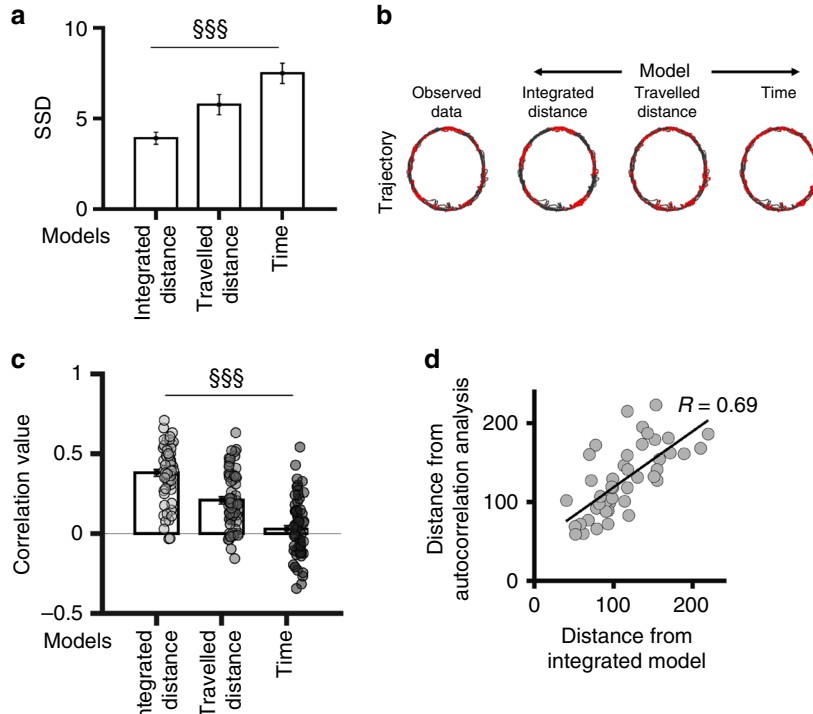

**Fig. 4** Grid cell regularity is based on path integrated distance-model fitting. **a** Average value (±SEM) of the fitting score (i.e. the sum of squared differences of the best fitting) for the path integrated distance, travelled distance and time model. §§§$p < 0.001$, repeated measure ANOVA. **b** Activity of one grid cell in the track from the real data (left) and reconstructed with best-fit parameters for each model (right 'MODEL'). Real (left) or reconstructed (right 'MODEL') trajectory of the animal with spike locations. **c** Average spatial correlation (±SEM) between the reconstructed maps from each model and the observed maps. Dots represent values from individual grid cells. §§§$p < 0.001$, repeated measure ANOVA. **d** Scatterplot showing the relation between the distance (i.e. path integrated distance) calculated with two different methods (i.e. autocorrelations and model fitting); the Pearson correlation coefficient 'R' is indicated

(Fig. 5d; one-way ANOVA: $F_{2129} = 44.34$, $p < 0.001$, Tukey HSD post hoc, large arena/small arena vs large arena/large track: $p < 0.001$, large arena/small arena vs large arena/small track: $p < 0.001$, large arena/small track vs large track/large small: $p < 0.001$). These results indicate that both the shape of the environment and its size profoundly influence grid cell metrics. A difference in the tactile input between the small and the large track (due to changes in the curvature of the walls) could potentially contribute to this process[27,28]. Next, we investigated whether the changes in field spacing were coherent across different environments (e.g. grid cells with a small field spacing in the arenas yielded also small firing distances in the tracks according to the integrated distance model). We found that field spacing was correlated

between all environments (Supplementary Figure 5e-g; large arena vs small arena: correlation coefficient = 0.71, $p < 0.001$; large arena vs large track: correlation coefficient = 0.48, $p < 0.001$; large track vs small track: correlation coefficient = 0.53, $p < 0.005$), indicating the maintenance of an internal coherence within the grid cell system across environments.

Altogether, our results demonstrate that in a continuous one-dimensional track, grid cell firing is influenced by a combination of dynamic idiothetic cues and static allocentric cues.

**MEC theta rhythm may underlie grid cell metrics changes.** LFP signals play a critical role in grid cell generation. In particular,

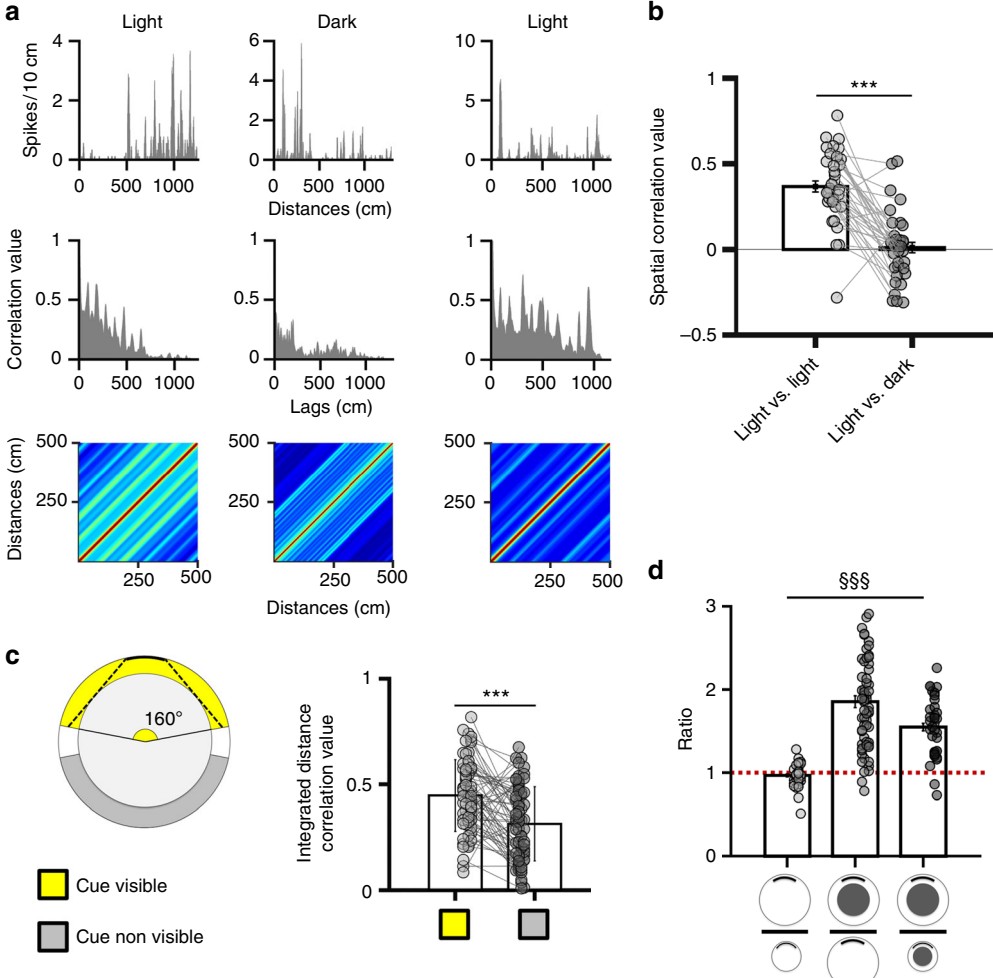

**Fig. 5** Environmental features influence grid cell activity. **a** Example of the activity of one grid cell linearized according to the path integrated distance model, in both light and dark conditions. Top row: firing rate; middle row: autocorrelation; bottom row: Toeplitz matrix. Note that the regularity is altered in darkness. **b** Average spatial correlation (±SEM) between the rate plots from all grid cells recorded in the two light sessions (light vs light) and in light vs dark sessions. Data from individual cells are shown in grey. ***$p < 0.001$, paired $t$-test. **c** Left: Diagram showing the two regions of the track in which the cue is is visible (yellow) or not (grey). Note that an ambiguous zone (white) was excluded from the analysis, so that the areas of the visible-cue and non-visible-cue region are equivalent. Right: Correlation (±SD) between the integrated distance firing activity in the visible-cue and non-visible-cue region of the track. Data from individual cells are shown in grey. ***$p < 0.001$ paired $t$-test. **d** Ratio (±SEM) between grid cell field distance in the arena and the track; the distance in the track is calculated based on the path integrated distance model. Dashed line indicates a ratio of 1 (i.e. same field distance in the two environments). §§§$p < 0.001$, repeated measure ANOVA. Note that grid cell field distance is unchanged between large and small arenas, whereas it significantly increases in the large track compared to all other environments

theta rhythm frequency has been pointed as a putative neural mechanism underlying grid cell scaling[29–31]. To examine whether possible changes in theta rhythm between the arena and the track could be responsible for the grid scale modifications we observed between the two environments, we analyzed grid cell firing rhythmicity and MEC local field potential (LFP) in the theta band. First, we found no difference in the average peak of firing theta modulation in the arena (8.83 Hz ± 0.58) and the track (8.72 Hz ± 0.51) (Fig. 6a, b; paired t-test: $t_{(63)} = 1.1$, n.s.). In contrast, analysis of LFPs in the theta band revealed a small but significant reduction of LFP-theta frequency in the track (7.82 Hz ± 0.32) compared to the arena (8.29 Hz ± 0.34) (Fig. 6c, d; paired $t$-test: $t_{(37)} = 6.21$, $p < 0.001$). The slope of the correlation between LFP theta frequency and running speed was also significantly decreased (track: 0.004 ± 0.006, arena: 0.01 ± 0.007; paired $t$-test $t_{(37)} = 4.68$, $p < 0.001$). These two modifications were not explainable by the reduction of running speed in the track (9.7 ± 0.1 cm/s) compared to the arena (14 ± 0.6 cm/s) (Fig. 6e; paired

$t$-test $t_{(37)} = -6.64$, $p < 0.001$). First, we observed no correlation between the changes in speed and LFP theta rhythmicity (Fig. 6f; correlation coefficient = −0.08, n.s.). Moreover, we recorded 23 speed cells[32] (N cells; rat 1 = 5, rat 2 = 11, rat 3 = 7) and found that the correlation between their firing rate and the animal speed was unmodified between the arena (0.64 ± 0.13) and the track (0.67 ± 0.13) (paired $t$-test: $t_{22} = −1.02$, n.s.). In contrast, changes in theta frequency significantly correlate with changes in field spacing between the two environments (coefficient correlation = 0.39, $p < 0.005$). Altogether these results highlight the LFP theta oscillations as a potential mechanism underlying changes within the grid cell metric system.

Finally, we observed no modifications in the firing properties of co-localized head-direction cells (Supplementary Figure 6; $N =$ 18). Their preferred firing direction (PFD) was stable between the arena and the track (Supplementary Figure 6d; average difference of PFD arena vs track: 2.35 ± 8.9°, Circular V-test with 0° expected mean direction changes: $v = 17.78$, $p < 0.001$), as

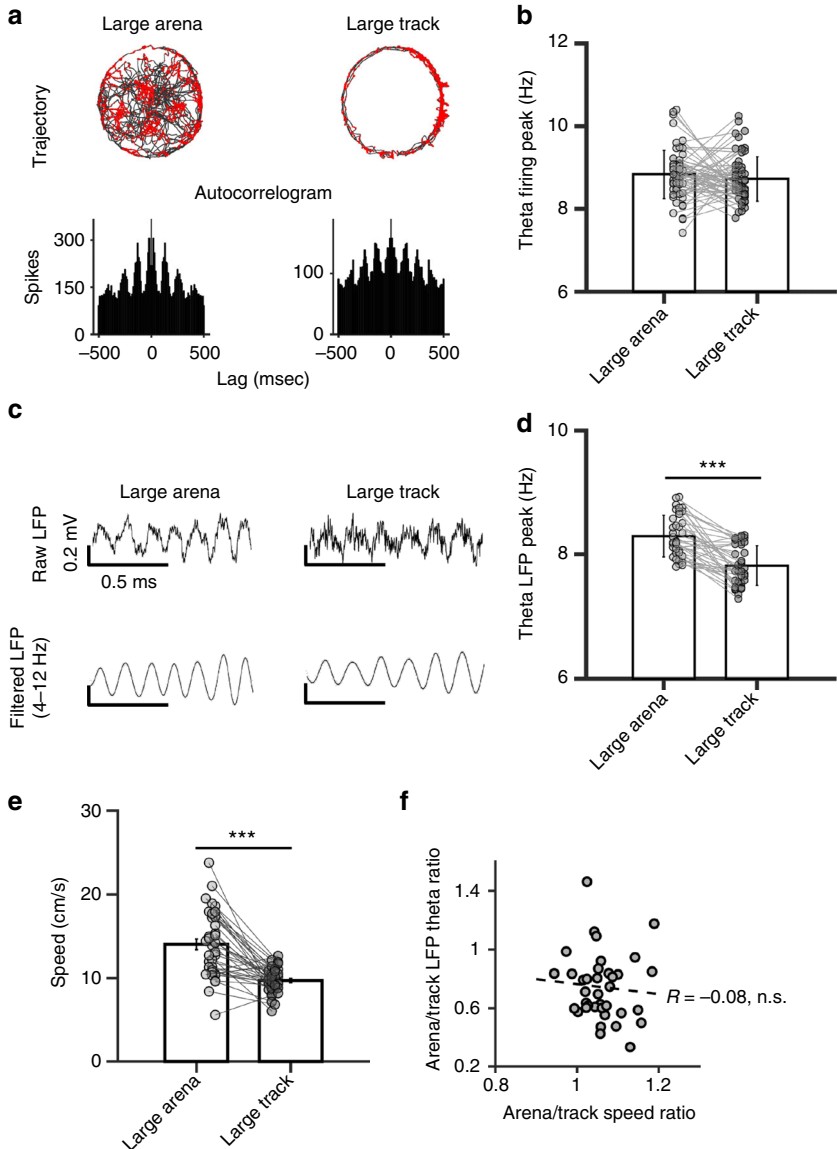

**Fig. 6** Theta rhythm in the track. **a** Autocorrelograms of the firing activity from one grid cell recorded in the arena and the track. Top row: trajectory with spikes locations. Bottom row: autocorrelogram of firing in 500 millisecond lags. **b** Average peak frequency (±SD) in the theta band (4–12 Hz) of the grid cell autocorrelogram. **c** Example of raw (top row) and filtered (4–12 Hz, bottom row) LFP recorded in the arena and the track. **d** Average peak frequency (±SD) in the theta band of LFPs recorded in the arena and the track. ***$p < 0.001$, paired $t$-test. **e** Average animal speed (±SEM) in the arena and the track. ***$p < 0.001$, paired $t$-test. **f** Scatterplot showing the relation between the arena/track speed ratio and the arena/track LFP theta peak ratio; the Pearson correlation coefficient 'R' is indicated with its significance (n.s.)

well as their directional selectivity (Supplementary Figure 6e; Rayleigh vector length arena = 0.59 ± 0.19; track = 0.58 ± 0.23; paired $t$-test: $t_{17} = 0.13$, n.s), peak firing rate (Supplementary Figure 6f; arena = 6.5 ± 6.2 Hz; track = 7 ± 4.9 Hz; paired $t$-test: $t_{17} = -0.96$, n.s), and mean firing rate (Supplementary Figure 6g; arena = 1.1 ± 1.3 Hz; track = 1.5 ± 1.1 Hz; paired $t$-test: $t_{17} = -1.37$, n.s).

## Discussion

According to previous studies, both self-motion cues and landmark information shape the grid cell firing patterns in 1D and 2D environments[26,33–38], thus raising the question of whether grid cell spatial activity is selectively determined by path integration. In our study we demonstrate that path integration is instrumental for grid cell spatial firing in a circular 1D environment. We show

that for most grid cells the distances between fields were constant across laps, thus indicating that spatial activity is relatively independent of the external reference frame and is consistent with a path integration mechanism. This linear pattern was neither transient, nor was it a consequence of the novelty of the environment since the rats were highly familiar with the track[25]. Interestingly, a smaller proportion of grid cells was controlled by allocentric coding rather than by path integration, thus suggesting that activity of some grid cells may be anchored to an external reference frame though this happens unfrequently.

Using the analytical approach of Yoon et al.[20], we demonstrated that the activity of grid cells in the circular track was not compatible with a slice of the 2D grid observed in the arena in contrast to others reports[18–20,39]. One explanation of this inconsistency is that our circular track had no physical discontinuity such as corners or U-turns compared to classical

corridors or other 1D environments previously employed, thus preventing resetting of the cellular estimate of animal position. Thus, in a circular track path integration mechanism determines the spatial firing pattern of grid cells.

We also show that grid cell regular activity in the circular track does not result from processing travelled distance or elapsed time, in contrast to the results of Kra[13]) who recorded grid cells as rats ran on a treadmill. In this study, allocentric information was kept constant while moving in the treadmill, so that rats could use neither allocentric distance nor path integrated distance (i.e. distance starting from a fixed reference point). Therefore, grid cell firing was driven by travelled distance and time, the only relevant information that was available to the animals. Our circular track allows to better disentangle the mechanism responsible for periodic grid cell firing since all information is available and possibly relevant for grid cell firing. Theoretically, a 1D or 2D pattern cannot result from the mere integration of travelled distance and time during a random foraging behavior. The variability of the animal path is such that the firing cannot be spatially organized. However, if the travelled distance or time component are referenced to a position, a stable spatial pattern could emerge.

Since path integration requires a starting point as a reference, and because the activity of most grid cells was not anchored to the allocentric space, the reference point cannot be within the external space. Rather field distance computational process appears to be related to an intrinsic mechanism of the grid cell system. One possibility could be that grid cells update distances not from a unique reference location (i.e. the starting position) but step by step by using each firing field as a reference to compute the distance to the next field.

Despite the prominence of path integration, grid cell activity in the track was also influenced to some extent by external cues. We showed that darkness significantly decreased the proportion of regular firing resulting from processing path-integrated distance. Moreover, firing fields were more coherent near the cue card. Landmarks may allow the animal to calibrate its position and reset errors[1,33,37] as shown in open arenas[34,35].

The external landmarks possibly exert an influence also on the metrical properties of the grid cell map. We observed an increase of the grid scale between the arena and the track that was permanent, while it was unmodified in arenas of different sizes. We hypothesize that this effect is the consequence of the reduced availability of the external information (i.e. the cue card) in the track compared to the arena. Since access to the visual cue is limited in the track, animals have to rely on self-motion cues to locate themselves. The resulting 1D map may be less accurate in representing the animal position over time due to error accumulation. If the firing fields are relatively close to each other, error accumulation during displacements may induce field overlapping, thus creating ambiguity about the animal position[40,41]. Increasing the distance between the fields therefore would reduce such ambiguity, as predicted from simulation of grid cell activity in environments with high spatial uncertainty, such as novel environments[41]. Field spacing increase would thus reflect the lower spatial resolution of the grid cell map following reduction in external cues. That grid scale was reduced in a smaller track, in which animals had a wider and more frequent visual access to the cue card, supports this view.

We finally analyzed MEC theta oscillations and cell firing oscillations as a possible mechanism underlying grid cell changes. We found a decrease in the frequency of MEC theta oscillations in the track, whereas no change in the cell intrinsic oscillations was observed. In addition, speed cell and head-direction cell activity were unmodified between the arena and the track. These findings suggest that theta oscillations are responsible for the metrical change of the grid map, thus supporting models

proposing that oscillatory brain activities with different frequencies are necessary for generating both the periodicity and the scale of the grid map[29,31]. Consistent with this result, reduction of theta rhythm by medial septum inactivation impaired both grid cell firing pattern and linear distance estimation in darkness[12,42,43].

Overall our results show that grid cells use a path integration process to compute a regular firing pattern in a continuous circular track. This is consistent with the hypothesis that grid cells are a key neural substrate for path integration[9,10]. However, this path integration process is not independent from the environment since the distance between firing fields is influenced by environmental features. It also depends on the temporal organization of the MEC local field potentials. Along with evidence of time and goal-distance coding in the hippocampus and the entorhinal cortex, our findings suggest that the entorhino-hippocampal system sets a representation of space by combining temporal, spatial, and motivational dimensions[14,44–48]. The integration of all these dimensions in a coherent representation may support our ability to remember events in space and time.

## Methods

**Animals and surgery**. Six Long-Evans male rats (Janvier, Le Genest-St-Isles, France) weighting 300–350 g were housed in individual cages (40 cm long x 26 cm wide x 16 cm high) with *ad lib* food and water and kept in a temperature-controlled room (20° + 2) with natural light/dark cycle. One week after arrival, animals were handled daily by the experimenter for 7 days. They were then placed on a food-deprivation schedule that kept them at 85–90% of their free-feeding body weight. During this period, they were habituated to the recording apparatus by letting them explore the different enclosures for 10–20 min per day. They were then implanted with tetrodes aimed at the medial entorhinal cortex (MEC) at the following coordinates: AP 0.4–0.6 mm in front of the sinus, ML 4.8–5.0 mm from the midline, and DV 1.5 mm under the dura. Surgery was performed under sterile conditions and general anesthesia (Ketamine 75 mg/kg (Imalgene 1000, Mérial, France)/Medetomidine 0.25 mg/kg (Domitor, Janssen, France). As a postoperative treatment, rats received an injection of antibiotic (Clamoxyl, 150 mg/kg) and of analgesic (Tolfedine, 4 mg/kg). After surgery, the rats were allowed 5–7 days of recovery.

**Microdrives and recording setup**. Each tetrode was composed of four twisted 25-μm nichrome wires. The four tetrodes formed a bundle threaded through a piece of stainless steel tubing. Each wire was attached to a pin on the outside of a rectangular Mill-max connector. The tubing was attached to the center pin of the connector and served as the animal ground as well as a guide for microwires[1]. The connector, tubing, and wires could be moved down by turning drive screw assemblies cemented to the skull. Before surgery, the wire tips were goldplated to reduce their impedance to 200–400 kΩ. Screening and recording were performed with a counterbalanced cable attached at one end to a commutator that allowed the rat to move freely. The other end of the cable was connected to the rat headstage, which contained a field effect transistor amplifier for each wire. The signals from each tetrode wire were further amplified 10,000 times, bandpass-filtered between 0.3 and 6 kHz with Neuralynx amplifiers (Neuralynx, Bozeman, MT, USA), digitized (32 kHz), and stored by DataWave Sciworks acquisition system (DataWave Technologies, Longmont, CO, USA). Two light-emitting diodes (LED), one red and one green, separated by 5 cm and attached to the headstage assembly provided the position and the orientation of the rat's head. The LEDs were imaged with a CCD camera fixed to the ceiling above the maze, and their position was tracked at 50 Hz with a digital spot-follower.

**Behavioral training and recording protocol**. Recording apparatus were four different black aluminum enclosures located in the same room and surrounded by black curtains: two circular arenas of 150 cm and 100 cm diameters and two circular tracks 15 cm wide same diameter, obtained by inserting opaque inner walls (same height as the arenas). All enclosures were polarized by a white cue card (30 cm width x 40 cm height) attached to the wall. The cue card was removed only for the recordings in total darkness. Randomly scattering sugar pellets from a food dispenser located above the enclosures motivated animals running. This procedure was used for both the arenas and the tracks. As a consequence, the rats did not run unidirectional laps in the tracks. They changed direction many times during exploration and rarely performed an entire lap, in both the large and the small track. Since a change in the running direction implies a 180° body turn, animals tended to slow down their running speed more often compared to the arena.

All the animals could explore the large arena and the large track every day for a period of at least 10 min, both before and after the surgery. Consequently, they

were highly familiar with the environments when the recording experiments started.

Two rats were exclusively trained in the two large environments according to the following procedure: two sessions in the arena and three sessions in the track (light–dark–light). In few recording experiments, the three sessions in the track were performed before the sessions in the arena. Since no differences were observed, all data were pooled together. Each recording session lasted 15–20 min. Between successive sessions, the rats were not disconnected but were removed from the apparatus and placed in a box in a random location outside the curtain for 2 min, to let the experimenter manipulate the apparatus (inserting the opaque inner walls, washing walls and the floor, removing the cue card, and switch off lights). Animals were then mildly disoriented by rotating the holding box before the next trial.

Four rats were recorded in the large arena, large track, small track, and small arena in light condition in random order. Each session lasted 15–20 min in the large environments, and 10–15 min in the small environments. The rats were removed from the enclosures between each session.

**Data analysis**. Spike sorting: Spike sorting was performed manually using the graphical cluster-cutting software Offline Sorter (Plexon). Units selected for analysis had to be well-discriminated cluster with spiking activity clearly dissociated from background noise. Units that were lost before the session series was completed, or whose waveforms changed too much between two sessions, were not used for further analysis. Units having interspike intervals <2 ms (refractory period) were removed due to poor isolation, as were cells with a peak firing rate ≤1 Hz. To prevent repeated recordings of the same cell over days, clusters that recurred on the same tetrodes in the same cluster space across recording sessions were only analyzed on the first day. A total of 365 cell clusters was accepted from which 64 were classified as grid cells based on their grid score (see below). Among them, 35 were recorded both in light and dark, and 34 were recorded in both large and small arenas and tracks.

Rate maps and autocorrelograms: Rate maps were first constructed by sorting the position data (estimated based on the red LED) into bins of 2.5 × 2.5 cm. The firing rate was determined for each bin and smoothed using the following algorithm:

$$\lambda(X) = \sum g\left(\frac{x_i - x}{h}\right) \Bigg/ \int_0^T g\left(\frac{y(t) - x}{h}\right) dt$$

where $g$ is a smoothing kernel, $h$ is a smoothing factor, $n$ is the number of spikes, $sj$ the location of the i-th spike, $y(t)$ the location of the rat at time t, and $[0, T]$ the period of the recording. A Gaussian kernel was used for $g$ and $h = 2 \times 2.5$ cm. Rate maps were constructed for arena and track recordings.

Spatial autocorrelograms of rate maps were used to assess the periodicity, regularity, and orientation of cells with multiple firing fields in open-fields[2,3]. Autocorrelograms were based on Pearson's product-moment correlation coefficient with corrections for edge effects and unvisited locations. With $\lambda(x,y)$ denoting the average rate of cells at location (x,y) the autocorrelation between fields with spatial lags of $tx$ and $ty$ was estimated as:

$$R(\tau_x, \tau_y) = \frac{n\sum \lambda(x,y)\lambda(x-\tau_x, y-\tau_y) - \sum \lambda(x,y) \sum(x-\tau_x, y-\tau_y)}{\sqrt{n\sum \lambda(x,y)^2 - \left(\sum \lambda(x,y)\right)^2} \sqrt{n\sum \lambda(x-\tau_x, y-\tau_y)^2 - \left(\sum \lambda(x-\tau_x, y-\tau_y)\right)^2}}$$

where the summation is overall $n$ pixels in $\lambda(x, y)$ for which the rate was estimated for both $\lambda(x, y)$ and $\lambda(x - t_x, y - t_y)$.

Grid scores: The degree of spatial periodicity (grid score) was determined for each recorded cell by selecting the six closest fields around the center of the autocorrelation map. The Pearson correlation was obtained between this map and its rotation at 60° and 120° degrees on one side and 30°, 90°, and 150° on the other. A grid score $g$ was defined as the difference between the minimum value in the first group and the maximum value in the second group[4]. We then used a shuffling procedure to classify the grid cells based on their 2D grid score[5]. We rotated the spike times with respect to the position vector by a random time drawn uniformly from the interval between 20 s and the recording duration minus 20 s. This procedure was repeated 400 times per cells (365 cells × 400 rotations = 146,000 rotations). For each rotation, we reconstructed the rate map and the autocorrelation map, and we calculated the grid score. We used the distribution of the scores from all rotations in all recorded cells to fix the statistical threshold (the ninety-fifth percentile of the distribution). All cells whose grid score was greater than or equal to this value were assigned as grid cells. The value of the ninety-fifth percentile was 0.32, similar to those reported in both rats[4,5] and bats[6]. 64 cells were categorized as grid cells and used for successive analysis.

Field spacing: Field spacing of the grid in the arena was defined as the average of the six distances from the central peak to the vertices of the inner hexagon in the 2D autocorrelogram[2].

Firing field variability: For each cell, we identified individual fields from which we extracted the peak firing rate, and we calculated the firing peak standard deviation. As for the track, we used the linearized path integrated firing activity (detailed methods are discussed below), and a field was identified if there were at

least five contiguous pixels (2.5 cm length) with activity greater than 30% of the cell peak firing rate. As for the arena, a field was selected from the spatial rate maps if there were at least nine contiguous pixels (2.5 cm) with activity greater than 30% of the cell peak firing rate. Firing field variability between the arena and the track was compared with a paired t-test.

**Allocentric and path integrated firing in the track**. Allocentric firing: we analyzed whether grid cells fire at the same position in the track across different laps for one given recording session. From the starting position of the rat, we calculated the angular travelled distance by the animal for each couple of x,y coordinates of rat trajectories, with positive distance when animal moves clockwise and negative distance when animal moves counterclockwise. Then we summed all distances samples. The circumference of the track being 471 cm (2.π.radius; radius = 75 cm), we dissociated each lap when cumulative distances reached a multiple of 471 cm. The allocentric activity was quantified by calculating the average Pearson correlation coefficient of cell activity between each lap.

Path integrated firing: we observed a shift of cell activity from one lap to the other. We analyzed whether this shift was consistent across laps. From cumulative polar distances extracted from the animal path as described in the previous section, we computed the spatial autocorrelation sequence. The value of the first peak was used as a measure of the distance between firing fields, for individual cells. For each cell, we used this distance to shift the firing sequence of each lap with respect to the firing sequence of the previous lap (Figure S2). If the cell continuously fires at a regular distance, the correction will consequently align the cell activity across laps. We calculated the average Pearson correlation coefficient of cell activity between each lap pairs, and we compared this with the average Z-transformed correlation coefficient of the allocentric firing.

Phase offset: We analyze whether firing associations of grid cells recorded simultaneously were maintained between the arena and the track. We found 13 recording sessions in which at least two grid cells were co-recorded (Rat1: two co-recorded grid cells and three co-recorded grid cells; Rat 2: two co-recorded grid cells and three co-recorded grid cells; Rat 3: three co-recorded grid cells and three co-recorded grid cells; Rat 4: three co-recorded grid cells and five co-recorded grid cells). For each pair of grid cells we calculated the 2D spatial cross-correlogram in the arena from which we extracted the distance offset as described in Fyhn et al[7]. In the track, we calculated the cross-correlogram of the path-integrated firing of co-recorded grid cells, and the distance of the first peak was used as a measure of phase offset. Finally, we calculated the Pearson correlation coefficient between the two offsets in the arena and the track, and we tested whether the Z-transformed value was significantly different from 0 using a one-sample t-test.

Grid cell directional firing in the track: For each cell we analyzed the path-integrated firing activity for clockwise and counterclockwise animal runs, and we calculated the Pearson correlation coefficient between the two firing sequences. A one-sample t-test was used to test whether the average Z-transformed correlation coefficient was different to 0.

Distance and time analyses in the track: To analyze whether cell firing was modulated by path-integrated distance, we calculated the angular distance as described in the previous section, then we averaged cell activity in 10 cm bins. We used a similar procedure for the travelled distance, except that we used absolute values of the distances (i.e. all angular distances, clockwise and counterclockwise, were positive). Then we summed all distances to compute total distance travelled by the animal and we averaged cell activity in 10 cm bins. Finally, for the time elapsed analysis, we used and summed the spike-time series for each cell, and we averaged firing activity in 1 s bins.

For each linearized firing activity according to three models (path integrated distance, travelled distance and time elapsed), we computed the autocorrelogram and the Toeplitz matrix of the autocorrelogram. Color-coded matrices are displayed for a distance range corresponding to roughly one lap of the track (500 × 500 cm).

To test whether grid cell activity was significantly modulated by path integrated distance, travelled distance, or time, we first calculated for each cell the average value of first two peaks of the autocorrelograms computed according to the three models. Then we compared each value with a control distribution obtained by using a "jitter" procedure. In this procedure we first added an amount of noise to the spike-time series (range of noise: from 1/5 of the smallest time difference between two spikes to 500 ms); then we computed the autocorrelogram of the linearized activity according to the three models; from the autocorrelograms we extracted the average correlation value of the first two peaks. This procedure was repeated 400 times per cell (365 cells × 400 rotations = 146,000 rotations). For each model (path-integrated distance, travelled distance, and time) we obtained a distribution of correlation values, which we used to set the statistical threshold as the ninety-ninth percentile (path-integrated distance: 0.18, travelled distance: 0.15, time: 0.14). All cells with average peak correlation values greater or equal to the statistical threshold were categorized as modulated by the corresponding coding model.

Model: Complementary to the previous analysis, we used a model framework to compare the time and distance modulation of spiking activity. For each cell, we re-arranged the recorded spikes to form gaussian firing fields that we distributed on the real animal path according to three parameters: (1) the width of spike trains, (2) the distance or time of the first spike train, and (3) the distance or time between spike trains. Model fitting score was assessed by computing the sum of squared

differences (SSD) between recorded (real) and modelled gaussian smoothed spike trains. For each model, best-fitting parameters were determined using a differential evolution algorithm implemented in MATLAB distributed under General Public License (http://www.icsi.berkeley.edu/storn/code.html)[9] minimizing the SSD. Based on these parameters, we built a rate map for each model and we computed the Pearson correlation coefficient between each modeled rate map and the experimental one. Z-transformed correlation values were compared with a repeated measure ANOVA and a Tukey HSD post hoc assessed differences.

Fourier spectral analysis: We analyzed whether grid cell activity in the track could be explained by a circular slice through a 2D triangular lattice. For each cell, we computed the spatial power spectral density (PSD) of its firing activity. If grid cell firing in the track corresponded to a slice of the 2D grid pattern, we should expect three peaks in the PSD[10]. In contrasts, if grid cell firing was regularly distributed in the track according to the path-integrated distance model, we should expect a single peak in the PSD. We therefore calculated a 'one peakness' score by dividing the area under the highest peak by the total area of the PSD, for each grid cell. As a control, we calculated the same score from the activity of each cell in a 'fake' track, reconstructed by superposing the path of the animal in the track to the extended 2D grid pattern. This procedure was performed several times by moving the path of the animal in order to cover all bins in a 2D triangular lattice. The ninety-fifth percentile of the distribution of all scores was used as statistical threshold for the real score (this analysis was performed on individual cells).

Small and large environments: We analyzed whether field spacing based on path-integrated distance model changed between small and large environments. For the 35 grid cells recorded in the large and small environments, we calculated the path-integrated distance modulation and tested its significance, as previously described. Then, we compared the proportion of grid cells modulated by path-integrated distance in the large and the small track, using a Chi-square test ($\chi^2$).

We calculated the Pearson product-moment correlation coefficient between (1) field spacing in the large arena and the path-integrated distance in the large track, (2) field spacing in the large and small arenas, and (3) path-integrated distances in the large and small tracks. Then, we calculated the ratio of distances between these three conditions that we compared using a repeated measure ANOVA and a Tukey HSD post hoc assessed differences.

Influence of the cue card: We tested the influence of the visual cue in the track using three analyses:

1. We recorded grid cells during successive light–dark–light sessions and calculated the field spacing based on the path integrated distance model. We compared the proportion of grid cells modulated by path integration in light and dark conditions using a Chi-square test ($\chi^2$). The Pearson correlation coefficient was also calculated between the two light sessions and between the first light and the dark sessions, for each grid cell. The two average Z-transformed correlation values were compared using a paired $t$-test.

2. We estimated the portion of the track in which the cue card was visible by calculating the intercept of the tangent to the inner circle of the track with a circle representing the trajectory of the animal (located in the middle of the track). The tangent origins from the border of the cue card. We found that in that condition an animal could have access to the cue card from an angle of 160° (out of 360°), which represents less than half of the track (Fig. 5c) Then, for each cell we computed the autocorrelogram of the path integrated firing in the region of the track in which the cue is visible. We ran the same analysis for the opposite region of the track in which the cue is not visible (Fig. 5c left panel). Then we extracted the correlation value of the first peak as a measure of firing stability. The average Z-transformed correlation values in the two regions of the track were compared using a paired $t$-test.

3. For each cell we estimated the density of firing fields located in the visible-cue and non-visible-cue regions of the track. A field was defined as least five contiguous pixels with activity greater than 30% of the cell peak firing rate. For each firing field we extracted the angular position. We then compared the proportion of firing fields identified in the visible-cue and non-visible-cue regions of the track to a 50:50 distribution using a Chi-square test ($\chi^2$). Finally, we compared mean firing rate from all grid cells in the two regions of the track using a paired $t$-test.

Speed cells analysis: The speed tuning of cells was assessed by correlating the cell firing rate with the rat's running speed. Cell firing rate was averaged across bins of 2 cm/s speed. Then we calculated the Z-transform of the Pearson correlation coefficient, the slope and intercept of the regression line, and tested whether they were significantly different from zero using a one-sample $t$-test.

Head-direction cells analysis: The rat's head direction was assessed for each tracked sample from the projection of the relative position of the two LEDs onto the horizontal plane. The directional tuning for each cell was obtained by dividing the number of spikes fired when the rat faced a direction (in bins of 6°) by the total amount of time the rat spent facing that direction. The peak firing rate was defined as the rate in the bin with the highest rate; the cell's preferred firing direction (PFD) was defined as the angle of this bin. We then used a shuffling procedure to classify the head-direction cells based on their directional specificity[8]. We rotated the spike times with respect to the position vector by a random time drawn uniformly from the interval between 20 s and the recording duration minus 20 s. This procedure was repeated 400 times per cells (365 cells × 400 rotations = 146,000 rotations). For each rotation, we reconstructed the directional tuning, and we calculated the

Rayleigh vector length. We used the distribution of the vector length values from all rotations in all recorded cells and we set the statistical threshold at the ninety-ninth percentile of the distribution, which was 0.24. All cells for which the Rayleigh vector length was greater or equal to 0.24 and for which the firing rate exceeded 1 Hz were classified as HD cells. We tested whether all 360° of compass heading were represented in the HD cell population by testing the non-uniformity of the distribution of PFDs with a Rayleigh test. We assessed the HD cell stability by calculating the angular difference in HD cell PFD between the arena and the track. A stable HD cell has an angular difference near 0°. We computed the distribution of angular differences that we tested using a circular v-test with a mean direction of 0°.

LFP-Theta rhythm analysis: Local field potentials (LFP) were recorded single-ended from one of the electrodes, simultaneously to single units. The signal was amplified 1000 times, lowpass-filtered at 500 Hz, sampled at 1024 Hz, and stored with the unit data. In order to characterize the oscillations in the theta band during exploration of the large arena and the large track, we bandpass the signal with cut-off frequencies of 4 Hz and 12 Hz (FMA Toolbox distributed under General Public License, http://fmatoolbox.sourceforge.net). We calculated the Hilbert transform of the resulting oscillations, and we extracted the instantaneous frequency (using hilbert. m function in Matlab), that was downsampled to the sampling rate of the animal position (25 Hz), to correlate with the instantaneous velocity of the animals. We then averaged the Z-transformed correlation values for all recordings and tested whether it was statistically different from zero using a one-sample $t$-test. Intrinsic firing frequency of grid cells in the theta band (4–12 Hz) was also calculated as the peak of the Hilbert transform of the cell spike-time autocorrelation sequence. Finally, we compared the average theta peak of LFPs or of cell intrinsic firing between the large arena and the large track using a paired $t$-test.

**Histology.** At the completion of the experiment, rats received an overdose of pentobarbital (Dolethal 10 ml/kg) and were perfused intracardially with 0.9% saline followed by 4% formaldehyde. The brains were removed, stored 1 day in formaldehyde, followed by a 30% sucrose solution, and finally frozen with dry ice. Thirty-micrometer-thick sagittal sections were mounted on glass slides and stained with cresyl violet. The positions of the tips of the electrodes were determined from digital pictures, acquired with a Leica Microscope (Wetzlar, Germany), and imported in an image manipulation program (Gimp 2.8, distributed under General Public License).

**Statistics.** Statistical tests included Student's $t$-tests, ANOVA, Pearson's correlations, Chi-square tests, Rayleigh tests, and circular V-tests. All Pearson correlation values had undergone a Fisher Z-transformation before statistical test

**Reporting Summary.** Further information on experimental design is available in the Nature Research Reporting Summary linked to this article.

## Data availability
The data that support the findings of this study are available from the corresponding author upon reasonable request.

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

## Acknowledgements
We thank Laure Spieser and Boris Burle for help with data analysis, and the Spatial Cognition group for their insightful comments on the manuscript. Financial support for this work was provided by the Institut Universitaire de France to F.S., Centre National de la Recherche Scientifique and Ministère de la Recherche. P.Y.J.'s salary was financed by the Fondation pour la Recherche Médicale (N° ARF20170938640).

## Author contributions
P.Y.J. and F.S. designed the study. P.Y.J. and F.C. performed surgeries and recordings. P.Y.J. and F.S. analyzed data. P.Y.J., F.C., B.P., E.S. and F.S. interpreted data and discussed results. P.Y.J. and F.S. wrote the manuscript. P.Y.J., F.C., B.P., E.S. and F.S. commented and edited the manuscript.

## Additional information

**Competing interests:** The authors declare no competing interests.

