## [Peer Review File · Nature Communications]

Reviewers' Comments:

Reviewer #1:

Remarks to the Author:

The manuscript by Jacob and colleagues investigates what spatial variables best predict the firing properties of grid cells as rats run on a continuous circular track.

They compared four variables: allocentric distance based on external landmarks, path integration (PI) distance, traveled distance and time. The central claim of the paper is that a PI process controls the firing activity of grid cells on the circular track. This process also strongly depends on visual information, as the activity of grid cells was impaired in darkness.

The results of the paper are interesting, and the methods used are up to the standards in the field. The observation that some firing fields appear to occur earlier on each lap is surprising and novel. While I am generally enthusiastic about the manuscript, additional analysis is needed to provide more support to some of the claims made in the manuscript.

Major comments:

One point that highlighted in the abstract is that the manuscript reveals a novel 1D equidistant firing pattern on the linear maze. Previous work by Ila Fiete (2016) showed that on a linear track, the firing of grid cells could be interpreted as a linear slice through a 2D lattice. They describe that the firing fields have non-periodic spacing, broad range of field heights, and the field spacing is much larger than in 2D environments. Is this the case on the circular track when using the path integration reference frame? It might also be useful to calculate power spectra of the firing rate in the PI reference frame (Yoon et al. 2016). If the fields are periodic, there should be only one peak in the power spectra. Such analysis would contrast the current findings with those of previous studies and would provide more evidence that the grid cell firing pattern on the circular track is genuinely novel.

Can the present findings be interpreted as a slice through a 2D lattice? The point of view of the authors is not entirely clear on the subject. They propose that "grid cells linearized their firing using a continuous PI process. What is meant by this? Are the results supporting the slice hypothesis or not?

Were some of the grid cells recorded simultaneously? It would be interesting to test whether the firing associations of the grid cells are maintained between the 1D and 2D environments. This information could help constrain the explanations of grid cell activity on the track. Based on previous work, we would expect stable firing associations between grid cells.

Previously published work suggests that the type of information processed by grid cells depends on what cues are available or important for the animal. For example, in open-field exploration with several landmarks, grid cells activity is dominated by allocentric distance and path integration. In this situation, time and distance traveled alone could not lead to grid fields, as acknowledged at line 226. However, when the animal runs on a treadmill, e.g., as part of a memory task, the allocentric information is held constant, and time and distance traveled control grid cell activity (Kraus et al., 2015). It might be useful to mention this point in the introduction (approximately at line 40). In this context, the current manuscript provides novel information about what information controls grid cell activity on a circular track. The main surprise is that the grid fields are not stable despite the presence of a polarizing cue and that the fields are periodic.

Please provide more information concerning the behavior of the animal. How many laps did the rats performed in each recording sessions? How much time did it take for an animal to complete one lap? How often did they change direction?

It is surprising to see clear firing fields in the 2D maps of the track. On the one hand, the rate maps of grid cells on the track show clear firing fields (Figure 2A and 2B). On the other hand, the authors suggest that the fields precessed from lap to lap, which should lead to very diffused firing fields in the 2D maps of the track. So why are there well-defined firing fields in Figure 2a and 2b? Perhaps it would be helpful to show several examples of maps like in Figure 3a (but also including the integrated distance).

What happens when the rats changed direction? Are the firing fields located at the same locations? In other words, are the firing fields uni- or bi-directional.

The procedure used to align each lap based on the average grid cell spacing is not clearly explained. What is the definition of a firing field? How is the mean field spacing used to align the laps? Perhaps illustrating the procedure in a supplementary figure would make the analysis clearer. This is very important as it is needed to convince the reader that a PI process drives grid cell activity. Are the firing fields on single laps equally spaced? If so this should probably be demonstrated before applying this procedure. How can the procedure be applied to cell `mec18.tet0.cell1`? This cell only has one firing field per lap. Is across-lap information used to realigned the laps? If so, this could artificially increase the correlation observed.

Line 81: The hypothesis that allocentric information is contributing to the firing of grid cells is ruled out prematurely. For example, are the firing fields more stable near the cue card? It seems possible that the cue card could help anchor some firing fields. The allocentric coding has a correlation of 0.13. This coefficient is low but is it significantly above chance? Were the fields unstable in all recording sessions or were there some stable fields in some sessions?

There is no mention of error accumulation in grid cell activity relative to integrated distance. For example, an animal could run two clockwise turns (from 0 to 942 cm), and then turn around to run two counterclockwise turns (from 942 to 0). If noise is accumulating in the grid cell network, the overlap between the fields should decrease as a function of time or distance traveled.

The results indicate a change in LFP theta frequency, with lower frequencies in the large track compared to the large arena. Running speed is known to influence theta frequency. It would be important to test whether a change in running speed explains the change in theta frequencies. Also, the reported theta frequencies are lower for spiking activity than for the LFP. Given that some grid cells show theta phase precession, one would expect higher frequencies for spiking activity than for LFP.

Minor comments:

The title could be improved by making it more specific. Perhaps including "in a 1D environment" in it or something similar.

Line 220: The authors suggest that their results are not consistent with those of Kraus et al. (2015). They suggest that distance traveled and time were confounded with path integration in this paper. However, Kraus manipulated the speed of the treadmill, which dissociated distance from time.

The threshold for grid scores and other spatial variables should be calculated on a cell-by-cell basis (Diehl et al. 2016).

Why are the peak firing rates in Figure 3b higher for integrated distance than for traveled distance? I

presume a different binning was used, but the method section states that the binning is 10 cm for both types of distance.

Please report the number of cells recorded in each animal.

"Rats had no access to the visual cue at least in half of the track." This estimate could be more precise.

Line 74: "In the track, grid cells fired at different locations during exploration." The meaning of this sentence is not immediately clear. Perhaps something like this would be better: "The position of firing fields in the track did not correspond to the position of the fields in the arena."

"Constrained in an endless environment" seems contradictory. Perhaps just "circular environment."

What are the "correlated events" on the y-axis of Figure 3b?

On Figure 3b, top row, first column. Are the data shown for the entire recording session? Also, why is the integrated distance always positive? If the animal ran more counterclockwise, shouldn't the values be negative? Also, why is the integrated distance for several Figures (3b,3c,5a) stops near 1200 cm?

Line 243: The authors suggest that the visual landmark allowed the animal to reset errors in PI. If this is the case, then the firing fields near the cue card should be more stable than the firing fields further away.

I encourage the authors to make the computer code used for the analysis available online to other scientists.

Reviewer #2:

Remarks to the Author:

In this paper, Yves-Jacob et al ask whether grid cells on a circular track respond to allocentric location, net distance traveled, total path length, or time. Surprisingly, they find that grid firing fields appear to better correlated to net distance traveled (which they call "path integrated distance") than allocentric location on the track. They show that these firing patterns are better explained by path integrated distance than by total path length or time. They have some data in which the lights were turned off, and show that grid patterns disorganize in the dark, in agreement with multiple previous reports. They find that grid spacing is larger on a 1.5m diameter circular track than in a corresponding 1.5m circular arena or on smaller (1m diam) tracks or arenas. Finally, they show that a small number of head direction cells appear to be consistent between the 1.5m diameter arena and the 1.5m track, but the peak frequency of theta oscillations decreases. They suggest this as an explanation for the expanded grid spacing on the 1.5m diameter track.

While the finding that on a circular track grid cells align better to path integrated distance than to allocentric location is interesting, the paper suffers from a number of issues that make it unfit for publication at this point. As a whole, the results present a confusing picture which is neither internally consistent nor consistent with the large body of literature that already exists on this topic. More experiments and analyses would be needed to address these issues, as explained below.

1) It is already known that grid cells rely on a combination of path integration signals and landmark input which serves to correct errors. Consistent with this, and in agreement with previous reports, grid

cells appear to disorganize when the lights are turned off—that is, their maps are uncorrelated with the lights-on condition and fewer cells are tuned to path integrated distance (though see number 6 below regarding this finding). How can this be reconciled with their finding that grid cells are uncorrelated with allocentric location on the track? What model could explain how a landmark input that occurs at a random phase of the grid cycle every lap could provide error correction?

2) The paper does not carefully consider all the available sensory signals and how they might be relevant in their task. They present a false dichotomy between path integration and external input, when in fact optic flow can be used to path integrate (Kautzky and Thurley 2016), and input from the whiskers could also be relevant (Chorev et al 2016, Sofroniew et al 2015), especially in narrow linear environments. Could the difference in spacing between the large and small circular tracks have to do with the curvature of the walls, which could influence whisker or tactile inputs? In the abstract, the authors state that “reducing access to visual information resulted in increased field spacing,” when it is not clear at all that this is the cause of the change. This part of the paper would be much more interesting if the authors could experimentally pinpoint the cause of this change, by more carefully controlling the available sensory cues.

3) Several findings disagree with published results in the field, but the authors present weak explanations for why this might be the case.

a. Kraus et al showed that grid cells respond to both distance and time traveled on a treadmill. The authors claim that “these two features were confounded with path integrated distances on the treadmill.” In fact, in that paper the speeds of the treadmill were set such that distance and time could be disambiguated. The authors need to do a better job of addressing the differences between their findings and those of Kraus et al. Were other MEC cells tuned to elapsed time?

b. Their grid fields appear to be uniformly spaced (but see number 6 below) whereas most other papers find unevenly spaced firing fields in linear environments (e.g. Domnisoru et al, Yoon et al, Perez Escobar et al, Eggink et al), suggestive of a “slice” through a 2D pattern. In these papers, field spacings are uneven within a single lap. Recordings in a long, landmark-free linear track would help clarify this discrepancy. If the authors are correct in their interpretation of the data, grid cells should have evenly-spaced firing fields in such an environment.

c. In multiple previous papers, grid cells rotate when the cue card is rotated, indicating cue-locking. Why is cue-locking not apparent in the data shown here?

d. In general, with such a large literature on this topic, the authors need to do a better job thinking about how their results fit in with this existing literature, what differences might be key between their experiments and previous ones, and how their findings move the field forward, beyond the already-known fact that grid cells are in part driven by path integration. The authors should address the fact that grid cell coding properties could change depending on task demands and features of the environment (such as number and salience of landmarks).

4) Some important details of the data were not presented.

a. How many laps were run on average in a session? It appears that a small number of laps were typically run (3-5). The data would be more convincing if the authors could record for longer period of time and show that the distance coding persisted over many laps.

b. How many grid cells were recorded per animal?

c. Supplementary fig 2, third cell: How can it be shown that grid fields are consistently spaced when there are only two laps and one field per lap? Was this a common occurrence in the data?

5) Many of the details of the analysis were unclear.

a. How exactly were grid cell firing fields shifted between trials to compute “integrated distance” correlations in Figure 2d? The amount that each lap is shifted should depend on the length of the track and the grid spacing, but this was not adequately explained in the Methods.

b. Most of the statistics were performed on z-scores. How were these z-scores computed?

c. Why were jitters of 0-500 ms used in Figure 3? Why not longer shifts?

d. There appears to be a disagreement between the statistics in the main text and Figure 2d (main text: $t_{63} = -2.22$, figure legend: $p < 0.001$).

6) The following claims do not have adequate support, at least from the analyses that are shown in the paper.

a. Grid cell firing fields are uniformly spaced. The authors could, for example, show a histogram of field-to-field spacings, plot field spacings vs. lap, or an FFT of the linearized rate map, to better support this claim.

b. Grid cells disorganize during the lights-off condition. More analysis should be done here. Are the fields more diffuse? Are the spacings between fields more variable?

7) The authors make a very weak logical jump from observing that MEC theta frequency decreases in the large circular track to claiming that this is responsible for the changes in grid spacing. At the very least, the following should be considered:

a. How do MEC speed cells respond to the circular track?

b. How does the slope of MEC theta change?

c. Does running speed of the animal change between environments?

8) It is unclear why head direction cells are relevant to distance calculations, or to this paper in general, yet they are presented anyway.

Reviewer 1: Major comments.

1) One point that highlighted in the abstract is that the manuscript reveals a novel 1D equidistant firing pattern on the linear maze. Previous work by Ila Fiete (2016) showed that on a linear track, the firing of grid cells could be interpreted as a linear slice through a 2D lattice. They describe that the firing fields have non-periodic spacing, broad range of field heights, and the field spacing is much larger than in 2D environments. Is this the case on the circular track when using the path integration reference frame? It might also be useful to calculate power spectra of the firing rate in the PI reference frame (Yoon et al. 2016). If the fields are periodic, there should be only one peak in the power spectra. Can the present findings be interpreted as a slice through a 2D lattice? The point of view of the authors is not entirely clear on the subject. They propose that “grid cells linearized their firing using a continuous PI process. What is meant by this? Are the results supporting the slice hypothesis or not?”

We show that grid cell firing from lap to lap corresponds to a 1-D pattern based on path integrated distance. We have changed the formulation accordingly (lines 11-13). Furthermore, we demonstrate that grid cell firing in the circular track does not reflect a slice through a 2D lattice using the following two analyses, adapted from Yoon et al. (2016), as suggested by the reviewer.

- 1- In the first analysis, for each grid cell we extracted the peak rate of each field in the arena and we used the peak rate standard deviation to estimate the range of the firing field heights. We did the same analysis on the path integrated firing in the track. A paired t-test showed no statistical difference between the track and the arena, inconsistent with the hypothesis of grid cell activity being interpreted as a circular slice through a 2D lattice. Figure 2c shows the result of this analysis. The main text (lines 97-98) and the supplementary text (lines 743-750) have been changed accordingly.
- 2- In the second analysis, we calculated the power spectral density (PSD) of the autocorrelogram of the grid cell firing in the path integrated reference frame. Then for each cell, we calculated a ‘one peakness’ score (surface under the peak divided by the total surface, adapted from Yoon et al. 2016). As a control, we calculated the same score from the activity of each cell in a ‘fake’ track, reconstructed by superposing the path of the animal in the track to the extended 2D grid pattern. This procedure was performed several times by moving the path of the animal in order to cover all bins in a 2D triangular lattice. The 95th percentile of the distribution of all scores was used as statistical threshold for the real score (analysis performed on individual cells). We observed that 50/64 (78%) of the cells presented a score above the threshold, indicating that the firing of the majority of grid cells in the track corresponds to a 1D regular pattern and is therefore not

explained by a circular slice through a 2D lattice. Figure S3 shows both the analysis and one example of grid cell. The main text (lines 98-101) and the supplementary text (lines 824-836) have been changed accordingly.

2) Were some of the grid cells recorded simultaneously? It would be interesting to test whether the firing associations of the grid cells are maintained between the 1D and 2D environments. This information could help constrain the explanations of grid cell activity on the track. Based on previous work, we would expect stable firing associations between grid cells.

We found 21 grid cells that were recorded simultaneously within 7 recording sessions in 4 animals (Rat1: 2 co-recorded grid cells and 3 co-recorded grid cells; Rat 2: 2 co-recorded grid cells and 3 co-recorded grid cells; Rat 3: 3 co-recorded grid cells; Rat 4: 3 co-recorded grid cells and 5 co-recorded grid cells). To test the presence of stable firing associations between the arena and the track, we calculated the phase shift between pairs of grid cells from their cross-correlogram (if there were more than 2 grid cells in the same recording session, we did the analysis for all pair combinations). For the arena we used the 2D spatial cross-correlogram, and for the track we used the path integrated firing cross-correlogram. Phase shift in the arena was quantified as the distance from the center to the closest peak (as described in Fyhn et al., 2007), whereas in the track it was the position of the first peak in the cross-correlogram (i.e. the distance from zero). The two shift values are significantly correlated (Pearson correlation coefficient $r=0.6025$, $p=0.0049$), indicating that firing associations are maintained between the arena and the track (Figure 2f, main text lines 131-137 and supplementary methods lines 771-781).

3) Previously published work suggests that the type of information processed by grid cells depends on what cues are available or important for the animal. For example, in open-field exploration with several landmarks, grid cells activity is dominated by allocentric distance and path integration. In this situation, time and distance traveled alone could not lead to grid fields, as acknowledged at line 226. However, when the animal runs on a treadmill, e.g., as part of a memory task, the allocentric information is held constant, and time and distance traveled control grid cell activity (Kraus et al., 2015). It might be useful to mention this point in the introduction (approximately at line 40). In this context, the current manuscript provides novel information about what information controls grid cell activity on a circular track. The main surprise is that the grid fields are not stable despite the presence of a polarizing cue and that the fields are periodic.

We agree with the reviewer and we have changed the text to better address this issue (lines 41-47).

4) Please provide more information concerning the behavior of the animal. How many laps did the rats performed in each recording sessions? How much time did it take for an animal to complete one lap? How often did they change direction?

We now incorporate this information in two additional tables. Table 1 summarizes the behavior of each implanted animal as requested by the reviewer. Table 2 summarizes the number of cells recorded from each animal. Explicit references to the tables are inserted in the main text (lines 74 and lines 82-83).

5) It is surprising to see clear firing fields in the 2D maps of the track. On the one hand, the rate maps of grid cells on the track show clear firing fields (Figure 2A and 2B). On the other hand, the authors suggest that the fields precessed from lap to lap, which should lead to very diffused firing fields in the 2D maps of the track. So why are there well-defined firing fields in Figure 2a and 2b? Perhaps it would be helpful to show several examples of maps like in Figure 3a (but also including the integrated distance).

We agree with the reviewer that the chosen examples were misleading. We now provide more representative examples in figure 2a-b and figure 3a to avoid any confusion for the reader. Note that we have also changed figure S4 to show that grid cells have discrete fields in the track for few laps (e.g., 1 or 2), but no clear field after 5 to 6 laps due to drift-based offset (Fig. S4, panel b, cells 1 & 2).

6) What happens when the rats changed direction? Are the firing fields located at the same locations? In other words, are the firing fields uni- or bi-directional.

Grid cell firing in the track was clearly bidirectional. To address this point, we separately constructed for each cell the path integrated firing maps according to the animal clockwise and counterclockwise directional trajectories. We calculated the correlation coefficient between the 2 maps and found an average correlation of 0.69125 ± 0.175946 . One sample t-test shows significant difference from 0; (paired t-test: $t_{(63)} = 31.43002$, $p < 0.001$) (main text lines 101-105 and supplementary methods lines 782-785).

7) The procedure used to align each lap based on the average grid cell spacing is not clearly explained. What is the definition of a firing field? How is the mean field spacing used to align the laps? Perhaps illustrating the procedure in a supplementary figure would make the analysis

clearer. This is very important as it is needed to convince the reader that a PI process drives grid cell activity. Are the firing fields on single laps equally spaced? If so this should probably be demonstrated before applying this procedure. How can the procedure be applied to cell mec18.tet0.cell1? This cell only has one firing field per lap. Is across-lap information used to realign the laps? If so, this could artificially increase the correlation observed.

As recommended by the reviewer, we now clarify the procedure used to align each lap in figure S3a. We hope that this new figure addresses reviewer comments adequately. The analysis was not based on identified firing fields but performed on the total firing activity. The distance used to align the activity across laps corresponds to the distance of the first peak of the spatial autocorrelogram, for each individual cell. The results show that firing fields are clearly aligned after distance correction for the 50/64 grid cells that are modulated by path integration processes. This is possible because field spacing is regular within each lap (see figure S3). In contrast, fields are not regularly distributed for the remaining grid cells whose activity may be anchored to an allocentric reference frame. We removed the example mec18.tet0.cell1 because it is misleading and is not representative of the grid cells we recorded. It is indeed the only example we have in our data with the rat performing only two laps and the cell showing only two firing fields. For the sake of clarity, we detailed the analysis in the figure presented at the end of the present document. In this example, the rat performed only two laps, although the total distance travelled covered much more (i.e. the rat was often changing direction). The grid cell was active in two firing fields, one per lap. The procedure used to align the laps is the same as described above. We first calculated the spatial autocorrelogram of the total firing activity across laps, and we used the distance of the peak to shift one lap with respect to the previous one.

8) The hypothesis that allocentric information is contributing to the firing of grid cells is ruled out prematurely. For example, are the firing fields more stable near the cue card? It seems possible that the cue card could help anchor some firing fields. (...) Were the fields unstable in all recording sessions or were there some stable fields in some sessions?

We agree with the reviewer and we think that this issue was not sufficiently discussed in the previous version of the manuscript. The point we wanted to stress is that grid cell firing in the track is mainly driven by path integration, and not that allocentric information is not contributing at all to grid cell activity. In fact, it does, as suggested by the drop in spatial selectivity observed in darkness. To better address this issue we performed additional analysis to investigate whether the cue card exerted

some influence on the grid cell firing in the track, as suggested by the reviewer. We also looked more carefully at the activity of the minority grid cells whose firing was not explained by path integration.

Influence of the cue card

For each cell we computed the autocorrelation of the path integrated firing in the region of the track in which the cue is visible. We ran the same analysis for the opposite region of the track in which the cue is not visible (figure 5c left panel). Then we extracted the correlation value of the first peak as a measure of firing stability. We found that the average correlation in the visible-cue region of the track (0.45 ± 0.17) was significantly greater than the average correlation in the non-visible-cue region of the track (0.31 ± 0.17) (paired t-test: $t_{(63)} = 4.817796$, $p < 0.001$). This result has been added in figure 5, in the main text lines 197-204 and the supplementary methods lines 856-866. Next, we asked whether well-defined fields were homogeneously distributed in the track. If cell firing is more stable close to the cue card and less stable far from the cue card, we should observe an asymmetry of field distribution. For each cell we determined field location from the path integrated firing. A field was defined as 5 contiguous bins (2.5 cm long) showing firing activity greater than 50% of the peak activity. The map in figure S5b shows the asymmetrical distribution of the firing fields from all grid cells recorded (highest density close to the cue card). Indeed, we found that the proportion of fields both in the visible-cue region (233 fields = 65%) and in the non-visible-cue region of the track (109 fields = 30%) were significantly different from a random 50:50 distribution ($\chi^2_{(1)} = 29.5395$, $p < 0.001$). It should be noted that no difference in the average firing activity was observed between the two regions of the track (cue visible = $1.45 \text{ Hz} \pm 0.94$; cue non visible = $1.3 \text{ Hz} \pm 0.77$, paired t-test: $t_{(63)} = 1.53$, n.s), indicating that the asymmetry in firing field distribution was not caused by a difference in cell discharge. These results were added in figure S5 and in the main text lines 205-209 and in the supplementary methods lines 867-873.

Altogether these new results, combined with those of the darkness experiment, indicate that the visual cue card exerts a strong influence on grid cell activity in the track.

Grid cells no modulated by path integration:

We found 16 grid cells whose activity was not explained by path integrated firing. We show an example cell in Figure 3f-g. To better characterize their activity, we discretized their firing on each lap (Figure 3h) and computed the correlation value between laps, similarly to the analysis described in figure 2d (allocentric model). We did the same analysis with the 48 grid cells whose activity was explained by a path integrated model. Figure 3i shows that the average correlation was significantly greater for the 16 cells that are not modulated by path integration (0.24 ± 0.14), compared to the 48 cells that are modulated by path integration (0.09 ± 1.7 ; paired t-test $t_{(63)} = 3.12954$, $p < 0.005$). This result suggests that a minority of grid cells presented an activity that was anchored to an allocentric reference frame (main text lines 155-160).

(...) The allocentric coding has a correlation of 0.13. This coefficient is low but is it significantly above chance?

The correlation value is indeed significantly different from zero (One sample t-test $t_{(63)} = 5.738096$, $p < 0.001$), suggesting that the allocentric coding contributes to the grid cell activity in the track. To better address this issue we performed the same analysis on individual grid cells. We found that 11 cells presented a significant correlation in the allocentric condition, whereas 50 cells were significantly correlated in the path integrated condition. A X^2 test comparison shows that the proportions are significantly different ($X^2_{(1)} = 47.639$, $p < 0.01$). It should be noted that the 11 allocentric-modulated cells are included in the 16 grid cells whose activity is not explained by a path integration model (see previous point). This result has been added in the main text (line 124-130).

9) There is no mention of error accumulation in grid cell activity relative to integrated distance. For example, an animal could run two clockwise turns (from 0 to 942 cm), and then turn around to run two counterclockwise turns (from 942 to 0). If noise is accumulating in the grid cell network, the overlap between the fields should decrease as a function of time or distance traveled.

We believe that the analyses and results shown in the previous section (influence of the cue card) partly answer this comment. To further explore this issue, we tested whether grid cell firing was consistent across time. For each cell, we calculated the integrated distance firing for the first half and the second half of the recording session, then we extracted the value of the first peak of the autocorrelograms of the two halves. We found no significant difference between the first half (0.39 ± 0.11) and the second half (0.33 ± 0.13) of the session (paired-t test: $t_{(63)} = 2.6214$, $p = 0.09$), indicating that the firing regularity/stability does not decrease over time. This result could be explained by the influence of the cue card, that could prevent error accumulation. This result has been added in the main text (lines 211-217).

10) The results indicate a change in LFP theta frequency, with lower frequencies in the large track compared to the large arena. Running speed is known to influence theta frequency. It would be important to test whether a change in running speed explains the change in theta frequencies. Also, the reported theta frequencies are lower for spiking activity than for the LFP. Given that some grid cells show theta phase precession, one would expect higher frequencies for spiking activity than for LFP.

We thank the reviewer for this comment and we re-analyzed LFP frequency and cell intrinsic firing with the method of Kropff et al. (2015) which uses instantaneous frequency to calculate theta peak

values. For cell intrinsic firing, we found new values for the arena ($8.83 \text{ Hz} \pm 0.58$) and track ($8.72 \text{ Hz} \pm 0.54$), which were not significantly different (Paired t-test, $t_{(63)} = 1.0971$, $p=0.27$). For the LFP, we found new values for the arena ($8.29 \text{ Hz} \pm 0.34$) and the track ($7.82 \text{ Hz} \pm 0.32$). Paired t-test shows significant difference (Paired t test: $t_{(37)} = 6.2064$, $p<0.001$). LFP frequencies values are lower than cell intrinsic spiking, and are thus consistent with phase precession, as noted by the reviewer.

Frequencies values and statistical results have been modified lines 256-257, and figures 6b and 6d have been corrected. These new results do not alter our conclusion that LFP theta frequency could underlie changes in grid cell metric in the track.

As suggested by the reviewer, we analyze animal speed in the arena ($15 \pm 3 \text{ cm/s}$) and the track ($11 \pm 1 \text{ cm/s}$), and found that animals move more slowly in the track (figure 6e, Paired t-test $t_{(37)} = 7.4430$, $p<0.001$). To test whether this difference was responsible for the modifications observed in LFP theta rhythmicity, we checked whether changes in speed were correlated with changes in LFP. We found no significant correlation (figure 6f, Pearson $r = -0.08$, $p=0.63$).

Altogether, our results indicate that while speed and theta were significantly decreased in the track, we found no evidence that the change in running speed explained the change in LFP frequency.

These results were added in Figure 6e and in the main text lines 260-272.

Reviewer1: Minor comments.

11) The title could be improved by making it more specific. Perhaps including “in a 1D environment” in it or something similar.

We have changed the title accordingly.

12) Line 220: The authors suggest that their results are not consistent with those of Kraus et al. (2015). They suggest that distance travelled, and time were confounded with path integration in this paper. However, Kraus manipulated the speed of the treadmill, which dissociated distance from time.

We clarified this point in the text (lines 307-311). What we meant is that distance travelled and path integration were confounded in the study of Kraus et al. (2015), not the time.

13) The threshold for grid scores and other spatial variables should be calculated on a cell-by-cell basis (Diehl et al. 2016).

We performed a cell-by-cell analysis as suggested by the reviewer and found 4 more grid cells for Rat 1 (N=26 instead of N=22). We found the same number of grid cells for the 5 other animals. We used the most conservative procedure and did not include the 4 additional grid cells in the final analysis. We hope the reviewer will agree.

14) Why are the peak firing rates in Figure 3b higher for integrated distance than for traveled distance? I presume a different binning was used, but the method section states that the binning is 10 cm for both types of distance.

We confirm that we used the same 10-cm binning for integrated distance and travelled distance. We calculated the number of spikes every 10 cm according to each distance hypothesis. In the travelled distance condition, we summed the absolute distances ran by the animal every 40 ms, but without taking into account the animal speed. In this condition, activity is lower than integrated distance because cell firing is inconsistent with travelled distance hypothesis.

15) Please report the number of cells recorded in each animal.

We added Table 2 that summarizes the number of cells recorded from each animal (lines 82-83).

16) "Rats had no access to the visual cue at least in half of the track." This estimate could be more precise.

We estimated the portion of the track in which the cue card was visible by calculating the intercept of the tangent to the inner circle of the track with a circle representing the trajectory of the animal (located in the middle of the track). The tangent originates from the border of the cue card. We found that in that condition an animal could have access to the cue card from an angle of 160° (out of 360°), which represents less than half of the track. We added Figure 5c for illustration. The main text (lines 77-78) and supplementary text (lines 856-861) have been changed accordingly.

Line 74: "In the track, grid cells fired at different locations during exploration." The meaning of this sentence is not immediately clear. Perhaps something like this would be better: "The position of firing fields in the track did not correspond to the position of the fields in the arena."

We change the sentence as suggested by the reviewer (lines 84-85).

17) “Constrained in an endless environment” seems contradictory. Perhaps just “circular environment.”

We have changed the lines 75-78 of the main text, and we have removed the sentence “Constrained in an endless environment”.

18) What are the “correlated events” on the y-axis of Figure 3b?

The correlated events represented the number of spikes correlated at each distance value. To avoid any confusion, we replaced “correlated events” with “correlation values” in the y-axis of figures 3b-3g-5a and S5d.

19) On Figure 3b, top row, first column. Are the data shown for the entire recording session? ...

Yes, data are shown for the entire recording session.

... Also, why is the integrated distance always positive? If the animal ran more counterclockwise, shouldn't the values be negative?

The reviewer is correct, we have negative values if animal ran counterclockwise. However, to simplify the figure we set the lower distance value to 0.

... Also, why is the integrated distance for several Figures (3b,3c,5a) stops near 1200 cm?

To have a clear view on cell activity, we used examples in which animal ran around 2.5 laps (1177 cm) in the path integrated condition. We believed that examples of cell activity with greater integrated distance would make difficult for the reader to visually assess our findings. We added this information in the figure legends.

20) Line 243: The authors suggest that the visual landmark allowed the animal to reset errors in PI. If this is the case, then the firing fields near the cue card should be more stable than the firing fields further away.

This is exactly what we observed (see above).

21) I encourage the authors to make the computer code used for the analysis available online to other scientists.

We added a section lines 383-385 to inform the reader that both codes and data are available upon request.

Reviewer 2: Major comments.

1) **It is already known that grid cells rely on a combination of path integration signals and landmark input which serves to correct errors. Consistent with this, and in agreement with previous reports, grid cells appear to disorganize when the lights are turned off—that is, their maps are uncorrelated with the lights-on condition and fewer cells are tuned to path integrated distance (though see number 6 below regarding this finding). How can this be reconciled with their finding that grid cells are uncorrelated with allocentric location on the track? What model could explain how a landmark input that occurs at a random phase of the grid cycle every lap could provide error correction?**

We agree with the reviewer and we think that this is a critical issue and it was not sufficiently clear in the previous version of the manuscript (see also points 8-9 of the responses to the reviewer 1). As pointed by the reviewer, previous studies have already shown that self-motion cues together with landmark information influence the grid firing pattern. Our results go beyond that assumption, showing that in a circular track path integration processes are instrumental for grid cell spatial selectivity since firing fields are uniformly distributed across different laps. However, the fact that firing fields do not have a stable position across laps does not mean that the allocentric information is not controlling at all the grid cell firing. We have re-analyzed the data and found that the allocentric information is indeed influencing their activity. First, we showed that a small proportion of grid cells (11/64) was not controlled by path integration process but showed greater allocentric coding (i.e. firing fields were stable across different laps) thus suggesting that grid cell activity may be anchored to an external reference frame though this happens infrequently. An example of allocentric-modulated cell is shown in figure 3f-h (lines 155-160 of the main text). Secondly, we showed that the cue card strongly influenced grid cell activity. We analyzed grid cell firing in two regions of the track in which the cue is visible or not (figure 5c, supporting methods lines 850-866). In visible-cue region of the track grid cells are more stable (figure S5a) and well-defined firing fields are more numerous compared to the non-visible-cue region (figure S5b, main text lines 197-217, supporting methods lines 867-873).

We are not aware of any computational model that could explain how a landmark input that occurs at a random phase of the grid cycle every lap could provide error correction. However, these new results, combined with those of the darkness experiment, indicate that the visual cue card exerts a strong influence on grid cell activity, possibly preventing error accumulation over time. Accordingly, we also showed that grid cell firing is stable between the two halves of the recording session (lines 210-217).

2) The paper does not carefully consider all the available sensory signals and how they might be relevant in their task. They present a false dichotomy between path integration and external input, when in fact optic flow can be used to path integrate (Kautzky and Thurley 2016), and input from the whiskers could also be relevant (Chorev et al 2016, Sofroniew et al 2015), especially in narrow linear environments. Could the difference in spacing between the large and small circular tracks have to do with the curvature of the walls, which could influence whisker or tactile inputs? In the abstract, the authors state that “reducing access to visual information resulted in increased field spacing,” when it is not clear at all that this is the cause of the change. This part of the paper would be much more interesting if the authors could experimentally pinpoint the cause of this change, by more carefully controlling the available sensory cues.

Our purpose was not to address the influence of all types of external sensory cues, but rather to disentangle the role of the two major sources of information for navigation, i.e. movement-related cues and visual cues. We agree that the distinction between purely internally-generated cues and external cues is sketchy, and that optic flow is a special case. Note however, that in our study the arena and the circular track were homogeneously black (except for the white cue card) without any tactile pattern. We believe that such properties strongly decrease possibility for the animal to use optical flow. This was made more explicit in the text (lines 56-58).

We agree with the reviewer that changes of curvature between the small and the large track could influence changes of grid cells metrics via tactile sensory system like the whisker. We added this hypothesis in the result section (lines 234-239).

3) Several findings disagree with published results in the field, but the authors present weak explanations for why this might be the case.

a. Kraus et al showed that grid cells respond to both distance and time traveled on a treadmill. The authors claim that “these two features were confounded with path integrated distances on the treadmill.” In fact, in that paper the speeds of the treadmill were set such that distance and time could be disambiguated. The authors need to do a better job of addressing the differences between their findings and those of Kraus et al. Were other MEC cells tuned to elapsed time?

We apologize for our confusing explanation. Our claim was that path integrated distance and travelled distance were confounded in Kraus et al., (2015) study, while time was well dissociated. We changed the main text accordingly (lines 307-311). We think that grid cells rely on the information that are available and relevant for their spatial activity. In the treadmill used by Kraus et al. (2015),

the allocentric information is held constant and is hence irrelevant for spatial computations. As a consequence, path integrated distance and travelled distance are confounded. In contrast time can be easily separated from distance (by imposing different speeds) and becomes highly relevant for the animal to compute distance travelled. In that condition, grid cells are tuned to time elapsed, and possibly used this information to perform spatial computations. In our study we used a circular track containing a prominent cue card, in which animals are free to move and change direction, thus crossing the same position repeatedly. This allows to better disentangle the relative weight of different coding processes since all information types are available and potentially relevant to compute distance. In this condition we found that the large majority of grid cells compute distances using a path integration process. We did not find any grid cells tuned to elapsed time in our study.

b. Their grid fields appear to be uniformly spaced (but see number 6 below) whereas most other papers find unevenly spaced firing fields in linear environments (e.g. Domnisoru et al, Yoon et al, Perez Escobar et al, Eggink et al), suggestive of a “slice” through a 2D pattern. In these papers, field spacings are uneven within a single lap. Recordings in a long, landmark-free linear track would help clarify this discrepancy. If the authors are correct in their interpretation of the data, grid cells should have evenly-spaced firing fields in such an environment.

We found uniformly spaced fields only in the integrated distance model whereas they are not regularly spaced in the allocentric model (unless the field spacing is a multiple-divider of the track circumference). In that regards our results are similar to those shown in previous studies using linear corridors, that mainly analyzed grid cell spatial activity in an allocentric reference frame. In contrast, our results drastically differ from previous studies since the activity that we observed in the track is not compatible with a slice through the 2D grid pattern. We performed an analysis adapted from Yoon et al. (2016) and found that grid cell activity in the circular track cannot be explained by a circular slice through a 2D lattice (see also point 6 below and point 1 of the responses to reviewer 1). First, we showed that the range of firing field heights was similar in the arena and the track (figure 2c, main text lines 97-98 and the supplementary text lines 743-750). Second, the power spectral density (PSD) of the autocorrelogram of the grid cell firing presented one dominant peak, that is compatible with a 1D regular pattern rather than a circular slice through a 2D lattice (figure S2, main text lines 91-101 and the supplementary text lines 824-836).

These results indicate that the grid cells exhibit different coding mechanisms depending on the information available in the environment, as discussed in the previous point. It is possible that linear corridors are coded as 2D spaces probably due to the presence of barriers that help anchored the activity to an allocentric frame. Using a very long corridor may help decreasing the influence of

external cues, although animals will still rely on the two ends of the corridor. In contrast to linear tracks, a circular environment has no barriers, and rats are submitted to an endless trajectory. This condition may favor path integration process to a larger extent than a linear corridor (even a very long one) could do. Our results show that grid cells are not simply modulated by idiothetic cues, but that a path integration process is responsible for the emerge of a specific spatial firing pattern. In that regards our results are novel and helps clarifying the mechanism responsible for their spatial activity.

c. In multiple previous papers, grid cells rotate when the cue card is rotated, indicating cue-locking. Why is cue-locking not apparent in the data shown here?

We haven't done rotations between sessions to test cue-locking. However, figure 5b show that between two light sessions grid cell path-integrated firing is well correlated, suggesting that grid cell activity was anchored to some external landmark. In addition, we show that the cue card exerts a strong influence on the grid cell activity, as detailed in the previous point. We hope these new results will convince the reviewer that grid cell activity was anchored to the prominent landmark.

d. In general, with such a large literature on this topic, the authors need to do a better job thinking about how their results fit in with this existing literature, what differences might be key between their experiments and previous ones, and how their findings move the field forward, beyond the already-known fact that grid cells are in part driven by path integration. The authors should address the fact that grid cell coding properties could change depending on task demands and features of the environment (such as number and salience of landmarks).

We agree with the reviewer and we have changed the main text accordingly (lines 41-47, 283-304, 307-311, 317-331).

4) Some important details of the data were not presented.

a. How many laps were run on average in a session? It appears that a small number of laps were typically run (3-5). The data would be more convincing if the authors could record for longer period of time and show that the distance coding persisted over many laps.

We added Table 1 to summarize the behavior for each implanted animal. We also inserted new example of grid cells in figure S4 to show that the offset of firing fields in maintained across several

laps (Fig. S4, panel b, cells 1 & 2, 5-6 laps). The figure also shows that discrete fields are visible over few laps (e.g., 1 or 2) but no clear field after 5 to 6 laps due to drift-based offset.

b. How many grid cells were recorded per animal?

We added Table 2 to summarize cell distribution for each recorded animal. Tables references have been inserted in the text line 74.

c. Supplementary fig 2, third cell: How can it be shown that grid fields are consistently spaced when there are only two laps and one field per lap? Was this a common occurrence in the data?

We removed the example mec18.tet0.cell1 because it is misleading and is not representative of the grid cells we recorded. It is indeed the only example we have in our data with the rat performing only two laps and the cell showing only two firing fields. For the sake of clarity, we detailed the analysis in the figure presented at the end of the present document. In this example, the rat performed only two laps, although the total distance travelled covered much more (i.e. the rat was often changing direction). The grid cell was active in two firing fields, one per lap. The procedure used to align the laps is the same as described above. We first calculated the spatial autocorrelogram of the firing activity across laps, and we used the distance of the peak to shift one lap with respect to the previous one (see also point 7 of the responses to reviewer 1).

5) Many of the details of the analysis were unclear.

a. How exactly were grid cell firing fields shifted between trials to compute “integrated distance” correlations in Figure 2d? The amount that each lap is shifted should depend on the length of the track and the grid spacing, but this was not adequately explained in the Methods.

We clarify this point by explaining the procedure use to align each lap in the figure S3a. We hope this new figure answers the reviewer's concern.

b. Most of the statistics were performed on z-scores. How were these z-scores computed?

We now specify in the supplementary methods (lines 919-922) that correlation coefficient values were normalized by a Fisher z-transform to run statistical test.

c. Why were jitters of 0-500 ms used in Figure 3? Why not longer shifts?

Our maximum jitter was 50% of the lowest firing rate (1Hz) for grid cell selection. We used this maximum value to add temporal noise that was relevant for a biological cell firing, as detailed in previous studies (Amarasingham, A., Harrison, M.T., Hatsopoulos, N.G., and Geman, S. (2012). Conditional modeling and the jitter method of spike resampling. *J. Neurophysiol.* 107, 517–531).

d. There appears to be a disagreement between the statistics in the main text and Figure 2d (main text: $t_{63} = -2.22$, figure legend: $p < 0.001$).

We thank the reviewer for pointing out this error. We have corrected the p value in the figure legend by “ $p < 0.05$ ”.

6) The following claims do not have adequate support, at least from the analyses that are shown in the paper.

a. Grid cell firing fields are uniformly spaced. The authors could, for example, show a histogram of field-to-field spacings, plot field spacings vs. lap, or an FFT of the linearized rate map, to better support this claim.

As suggested by the reviewer, we adapted the analysis from Yoon et al. (2016) study to test whether grid cell firing was consistent with circular slice through 2D pattern, as discussed in the previous point. We computed the power spectral density (PSD) of the firing activity linearized according to the path integrated distance model, and we calculated a ‘one peakness score’ as the surface of the area below the highest peak of the PSD, divided by the total surface. The statistical significance of the score of each individual cell was tested using a bootstrap procedure, as detailed in the main text (lines 91-101). We found that 78% of the cells presented a single peak in the PSD. An example is shown in figure S2b. This result indicates that for the majority of the grid cells their firing activity is regularly spaced from lap to lap, consistent with a path integration process.

b. Grid cells disorganize during the lights-off condition. More analysis should be done here. Are the fields more diffuse? Are the spacings between fields more variable?

We ran additional analysis in the lights off condition but were not able to clearly identify firing fields since spatial activity was strongly disrupted. Based on this result we cannot conclude whether fields were more diffuse or spacing was more variable. Together with the new results showing a strong influence of the cue card on the grid cell spatial selectivity (discussed in point 1 above), our results

show that the availability of external landmarks deeply influence grid cell ability to show precise spatial code.

7) The authors make a very weak logical jump from observing that MEC theta frequency decreases in the large circular track to claiming that this is responsible for the changes in grid spacing. At the very least, the following should be considered:

a. How do MEC speed cells respond to the circular track?

We found 23 speed cells in 3 animals by analyzing their activity in the arena (Speed cell per rat: rat 1= 5, rat 2= 11, rat 3= 7). We compared speed correlation value, speed slope and intercept for the 23 speed cells, and we did not find any statistical differences.

- speed correlation: arena = 0.64 ± 0.13 , track = 0.67 ± 0.13 , $t(22) = -1.0167$, $p = 0.32$
- speed slope: arena = 0.063 ± 0.04 , track = 0.086 ± 0.06 , $t(22) = -1.6093$, $p = 0.12$
- intercept: arena = 1.85 ± 1.8 , track = 1.92 ± 1.67 , $t(22) = -0.1433$, $p = 0.88$

Methods for speed cells have been inserted lines 874-878.

These results indicate that the speed cell properties are highly preserved in the two environments. The comparison of the speed and cell firing correlations between the arena and the track has been inserted in the text lines 264-267.

b. How does the slope of MEC theta change?

We found a significant decrease of speed/theta correlation slopes between the arena (0.01 ± 0.007) and the track (0.004 ± 0.006); Paired t-test $t_{(37)} = 4.679553$, $p=0.00004$. This result has been added lines 260-262. Methods have been inserted lines 904-908.

c. Does running speed of the animal change between environments?

We analyze animal speed in the arena (15 ± 3 cm/s) and the track (11 ± 1 cm/s), and found significant differences (Paired t-test $t_{(37)} = 7.4430$, $p < 0.001$) indicating that animals run slower in the track (Figure 6e and lines 262-264). We tested whether changes in speed was correlated with changes in LFP, and we found no significant correlation (correlation value = -0.08 , $p=0.63$ figure 6f and lines 264-265 of the main text and lines 904-908 in the supplementary methods). Altogether, these results indicate that while speed and theta were significantly decreased in the track, there was no evidence that changes in running speed explain changes in LFP frequency.

8) It is unclear why head direction cells are relevant to distance calculations, or to this paper in general, yet they are presented anyway.

In our study, we observed changes in grid cell metric between the arena and the track. Previous studies showed that cell intrinsic firing (Giocomo et al., 2007), LFP theta frequency (Brandon et al., 2011; Koenig et al., 2011) and the anterior thalamic nucleus (Winter et al., 2015), an important structure for the head direction cell network, are important for the emergence of the grid cell pattern and its metric/scale. This is the reason why we think that analyzing whether all these biological activities (cell intrinsic firing, LFP and head direction cells) were changed between the arena and the track could help understanding the mechanism responsible for the modifications in grid cells activity we observed in this study. We agree with the reviewer that results of the head-direction cells are not the most important, and we moved them in supplementary figure 6.

a

Trajectory X spikes

b

1- Cell firing is dissociated for each lap

2- Cell firing autocorrelation

3- Simulated cell firing

4- Cell firing distance rescaling

Reviewers' Comments:

Reviewer #1:

Remarks to the Author:

The manuscript describes the activity of grid cells when rats explore a circular track. The main finding of the study is that grid cells still exhibit periodic firing in the track and that the locations of the firing fields depend largely on path integration mechanisms. The revised manuscript better acknowledges a modest contribution of allocentric information. Overall, the manuscript points to complex interactions between idiothetic and allocentric cues to control the activity of grid cells on a circular track.

The authors have done a good job answering most of the points I had raised in the first round of the review process. I still have a few comments on the manuscript that the authors might want to consider.

Major points

Line 112: The authors have clarified their method used to align the firing fields of different trials. I still have a question regarding the comparison of spatial correlation between path integrated distance and allocentric distance (Figure 2e). I suspect that there might be a bias in this analysis. In the case of path integrated distance, the spatial autocorrelation is used to calculate the offset between laps (distance b in Supplementary figure 3). If I understand the analysis correctly, the same data set is used to calculate the spatial autocorrelation and to calculate the path integrated distance correlation (Figure 2e). Could this procedure lead to an artificially high correlation? Could it be possible to split the data into a training set to calculate the spatial autocorrelation and a test set to calculate the path integrated distance correlation?

Line 292: A small group of grid cells encoded allocentric distance. Perhaps a sentence could be added to indicate whether grid cells encoding allocentric distance were recorded at the same time as grid cells encoding path integration distance.

Minor points

Line 34: "Distance can be calculated using either external cues, path integration processes or measuring time elapsed. Accordingly, four types of distances..." There are three elements listed in the first sentence but four in the second. Because the sentence starts with "Accordingly," I would expect three elements in the second sentence.

The names used for the four types of distances should be kept constant. I would suggest to list them on line 36 and always use the same terms throughout. For example, use "path integration distance" and not "distance based on path integration mechanisms."

Line 56: "The circular wall of the track..." I think there are two walls, an internal and external wall. Was the white cue card located on the external wall?

Line 91: "This activity." Perhaps change this to "Grid cell activity in the track" to be more specific.

Line 303: "In this condition": please spell out which condition this is.

Line 73: The sentence seems to suggest that the same cell was recorded from 6 rats.

Reviewer #2:

Remarks to the Author:

I feel the authors have adequately addressed my concerns.

Reviewer 1: Major points.

1) The authors have clarified their method used to align the firing fields of different trials. I still have a question regarding the comparison of spatial correlation between path integrated distance and allocentric distance (Figure 2e). I suspect that there might be a bias in this analysis. In the case of path integrated distance, the spatial autocorrelation is used to calculate the offset between laps (distance b in Supplementary figure 3). If I understand the analysis correctly, the same data set is used to calculate the spatial autocorrelation and to calculate the path integrated distance correlation (Figure 2e). Could this procedure lead to an artificially high correlation? Could it be possible to split the data into a training set to calculate the spatial autocorrelation and a test set to calculate the path integrated distance correlation?

We perform the analysis proposed by the reviewer to unravel any possible bias. This is indeed a crucial issue, since it could potentially invalidate one important result from our study. First, we selected recording sessions in which the animal ran at least 4 laps in order to have enough sampling for this analysis. 28 cells are included in the analysis (44% of the entire population). Then we splitted each session into two periods with the same number of laps, we calculated the spatial autocorrelation from the first period and we extracted the distance value, as previously described. This distance value was used to shift the spatial activity of the laps performed in the second half of the session, in order to calculate the spatial correlation between laps. From the second half of the session we also calculated the spatial correlation between laps without any shift, according to the allocentric model. The result of this new analysis confirmed the one previously reported: the average spatial correlation value was significantly greater after the path integrated distance correction than in pure allocentric coding (allocentric spatial correlation= 0.02 ± 0.01 , path integrated distance spatial correlation= 0.25 ± 0.04 , paired t-test on Z scores: $t_{28} = -5.89780$, $p < 0.000003$). The figure below shows the average correlation values as well as the individual values. This result has been added in the main text lines 125-131.

2) **A small group of grid cells encoded allocentric distance. Perhaps a sentence could be added to indicate whether grid cells encoding allocentric distance were recorded at the same time as grid cells encoding path integration distance.**

As proposed by the reviewer, we added a sentence in the results lines 145-148

“It should be noted that grid cells controlled by allocentric distance and grid cells controlled by path integrated distance were never recorded simultaneously. However, given the limited number of “allocentric” grid cells it is difficult to draw any firm conclusion from this observation.”

Reviewer 1: Minor points.

3) **Line 34: “Distance can be calculated using either external cues, path integration processes or measuring time elapsed. Accordingly, four types of distances...” There are three elements listed in the first sentence but four in the second. Because the sentence starts with “Accordingly,” I would expect three elements in the second sentence.**

We have changed the sentence lines 34-35

4) **The names used for the four types of distances should be kept constant. I would suggest to list them on line 36 and always use the same terms throughout. For example, use “path integration distance” and not “distance based on path integration mechanisms.”**

We have changed accordingly to the reviewer comment.

5) **Line 56: “The circular wall of the track...” I think there are two walls, an internal and external wall. Was the white cue card located on the external wall?**

We added “*attached to the external wall*” in the main text line 58.

6) **Line 91: “This activity.” Perhaps change this to “Grid cell activity in the track” to be more specific.**

We have changed accordingly to the reviewer comment in the main text line 92.

7) **Line 303: “In this condition”: please spell out which condition this is.**

We have changed by “In this condition” by “*Thus, in circular track*” lines 313.

8) **Line 73: The sentence seems to suggest that the same cell was recorded from 6 rats.**

We have changed “... grid cells recorded from 6 rats” by “*grid cells recorded across 6 rats*” lines 74.

Source data

A section Source Data has been added with the sentence “*Source data are provided as a source Data File*” line 390-391.